# Mesenchymal stem cells derived from patients with premature aging syndromes display hallmarks of physiological aging

Jean Philippe Trani[1], Raphaël Chevalier[1], Leslie Caron[1], Claire El Yazidi[1], Natacha Broucqsault[1], Léa Toury[1], Morgane Thomas[1], Karima Annab[1], Bernard Binetruy[1], Annachiara De Sandre-Giovannoli[1,2,3], Nicolas Levy[1,2,3], Frédérique Magdinier[1], Jérôme D Robin[1]

Progeroid syndromes are rare genetic diseases with most of autosomal dominant transmission, the prevalence of which is less than 1/10,000,000. These syndromes caused by mutations in the *LMNA* gene encoding A-type lamins belong to a group of disorders called laminopathies. Lamins are implicated in the architecture and function of the nucleus and chromatin. Patients affected with progeroid laminopathies display accelerated aging of mesenchymal stem cells (MSCs)–derived tissues associated with nuclear morphological abnormalities. To identify pathways altered in progeroid patients' MSCs, we used induced pluripotent stem cells (hiPSCs) from patients affected with classical Hutchinson–Gilford progeria syndrome (HGPS, c.1824C>T—p.G608G), HGPS-like syndrome (HGPS-L; c.1868C>G—p.T623S) associated with farnesylated prelamin A accumulation, or atypical progeroid syndromes (APS; homozygous c.1583C> T—p.T528M; heterozygous c.1762T>C—p.C588R; compound heterozygous c.1583C>T and c.1619T>C—p.T528M and p.M540T) without progerin accumulation. By comparative analysis of the transcriptome and methylome of hiPSC-derived MSCs, we found that patient's MSCs display specific DNA methylation patterns and modulated transcription at early stages of differentiation. We further explored selected biological processes deregulated in the presence of *LMNA* variants and confirmed alterations of age-related pathways during MSC differentiation. In particular, we report the presence of an altered mitochondrial pattern; an increased response to double-strand DNA damage; and telomere erosion in HGPS, HGPS-L, and APS MSCs, suggesting converging pathways, independent of progerin accumulation, but a distinct DNA methylation profile in HGPS and HGPS-L compared with APS cells.

## Introduction

Laminopathies define a heterogeneous group of diseases linked to mutations in the *LMNA* gene. Among these, Hutchinson–Gilford progeria syndrome (HGPS) characterized by premature aging in early childhood represents the most striking clinical condition (1, 2, 3, 4). Symptoms associated with this extremely serious rare congenital disease include growth retardation, alopecia, lipodystrophy, osteoporosis, osteolysis, thin skin, and loss of subcutaneous fat. The disease is also characterized by atherosclerosis and myocardial infarction, which is the leading cause of death in HGPS patients who die in early adolescence (5). This disease is caused by an autosomal dominant mutation in the *LMNA* gene encoding A-type lamins (6, 7). The mutation (NM_170707.4: c.1824C>T; p.G608G) consists in a C-T transition at position 1824 in exon 11. This transition activates an alternative splice donor site, leading to the production of a truncated prelamin A protein called progerin with a deletion of 50 amino acids in its globular C-terminal domain. Thus, in HGPS patients, progerin retains a C-terminal farnesyl tail, which is normally cleaved during posttranslational maturation of wild-type prelamin A.

At the cellular level, HGPS pathogenesis has been linked to a variety of causes, associated with the accumulation of progerin, which is toxic to cells, but also the fragility and disruption of the nuclear *lamina* resulting in nuclear blebbings, chromatin reorganization, premature cell senescence, increased sensibility to mechanical stress, and defects in autophagy (8, 9, 10). HGPS has also been associated to the depletion of the stem cell pool (11) and defects in the terminal differentiation of cells derived from the mesenchymal stem cells (MSC) lineage (12, 13, 14), such as adipocytes, chondrocytes, and osteocytes along with the loss of vascular smooth muscle cells contributing to atherosclerosis.

[1]Aix Marseille Univ, MMG, Marseille Medical Genetics U1251, Marseille, France    [2]Assistance Publique Hôpitaux de Marseille (APHM), Département de Génétique Médicale, Hôpital d'Enfants de la Timone, Marseille, France   [3]Biological Resource Center (CRB-TAC), APHM, La Timone Children's Hospital, Marseille, France

Correspondence: Frederique.Magdinier@univ-amu.fr; Jerome.robin@univ-amu.fr

Since the identification of the recurrent p.G608G HGPS-causing mutation, additional mutations in the *LMNA* gene have been associated with other segmental premature aging syndromes, with patients harboring symptoms closely related to classical HGPS. In 2007, a heterozygous mutation in the *LMNA* gene (p.T623S) was found in an 11-yr-old girl presenting with a phenotype similar to HGPS (15), including facial dysmorphism, alopecia, low bone mass, cardiovascular defects, increased triglyceride, LDL, and cholesterol levels. A patient previously described, who died at the age of 45 years from myocardial infarction, carried the same mutation (c.1868C>G) in exon 11 of the *LMNA* gene as the typical HGPS mutation (16). As in HGPS, this mutation creates a donor splice site or alternatively leads to modification of the protein sequence (p.T623S). After maturation of the protein, this cryptic splice site results in the deletion of 105 nucleotides from the messenger RNA and loss of 35 amino acids (lamin Δ35), including the excision site of the farnesyl group. The consequence is the same as in HGPS, with persistence of a toxic farnesyl group at the truncated prelamin A C-terminal tail (16, 17).

In addition, different syndromes called atypical progeroid syndromes (APS) because of lamin A/C missense mutations and characterized by a clinical phenotype similar to that of HGPS have been described. Hence, from a molecular point of view, the mutation spectrum in APS is different from classical HGPS with different variants reported, including a homozygous *LMNA* mutation (c.1583C>T; p.T528M) found in three boys presenting a mandibuloacral dysplasia (MAD) phenotype (A De Sandre-Giovannoli, personal communication); a composite heterozygous mutation (c.1583C>T and c.1619T>C; p.T528M and p.M540T, respectively) (18) and a heterozygous mutation c.1762T>C (p.C588R) (19, 20). Contrary to HGPS or HGPS-like, all these mutations neither modify the mechanism of protein maturation nor lead to progerin/prelamin A accumulation. In addition to signs of classical progeria, the patient developed a myopathy and MAD, both associated with other *LMNA* mutations (e.g., homozygous missense mutation c.1579C > T, p.R527C) (21).

Because mutations in A-type lamins associated with premature aging syndromes mainly affect tissues derived from MSCs, we sought to identify pathways that are dysregulated at early differentiation stages. To this aim, we produced induced pluripotent stem cells (hiPSCs) from patients affected with premature aging syndromes linked to different *LMNA* mutations (HGPS, n = 3; HGPS-L, n = 1; and APS, n = 3), established a differentiation protocol to obtain MSCs, and analyzed their transcriptome and methylome in comparison to cells from a healthy young donor and a healthy aged donor, used as controls. Importantly, hiPSCs hold the potential to produce standardized cell preparation (22) with large capacity of culture expansion. Thus, we produced hiPSC-derived MSCs and focused on events associated with early *LMNA* expression (e.g., lamins and progerin), including transcription, DNA methylation, and major cellular hallmark of aging (DNA damage and mitochondrial network). HIPSCs derived from these three laminopathy subgroups were indistinguishable from one another and from controls. However, upon MSCs differentiation, all cells displayed signs associated with cellular aging. We found that APS MSCs displayed differences in methylated DNA and transcription profiles when compared with HGPS and HGPS-L cells, indicating specific

molecular mechanisms depending on the presence or absence of farnesylated prelamin A/progerin but leading to a similar premature aging phenotype.

# Results

## MSC differentiation is not impaired in cells from patients affected with premature aging

HGPS and associated syndromes (HGPS-L and APS) are characterized by defects in specific tissues, especially those derived from the mesenchymal cell lineage (adipose tissue, bones, cartilage). To uncover genes and pathways associated with MSCs premature aging in patients with mutation in the *LMNA* gene, we generated hiPSCs from different patients and healthy donors. HiPSCs were derived from three patients affected with HGPS (c.1824C>T, p.G608G); one patient with a progeria-like syndrome, HGPS-L (c.1868C>G—p.T623S) and three patients affected with APS (homozygous c.1583C>T, p.T528M; heterozygous c.1762T>C p.C588R; compound heterozygous p.T528M and p.M540T, respectively) (Fig 1A and Table S1). As controls, we used hiPSCs derived from a young healthy donor (matched in age with the different patients) and hiPSCs derived from an 82-yr-old healthy donor. All hiPSCs were characterized following standard procedures (23, 24). Upon reprogramming, hiPSCs expressed the classical pluripotency markers (*SOX2*, *NANOG*, *OCT4*; Fig S1A), form embryonic bodies expressing markers of the different embryonic layers (Fig S1B), and displayed a normal karyotype (data not shown). To note, although HGPS-hiPSC lines expressed lower levels of pluripotency markers, we did not observe any differentiation defects in these lines. As expected, A-type lamins and progerin were not produced in stem cells (12, 25) (Fig S1C).

After optimization of the mesenchymal cell lineage differentiation from hiPSCs, hiPSC-derived MSCs, dubbed thereafter MSCs (26) (Fig 1B), we determined the percentage of cells expressing MSCs markers at different passages by flow cytometry using the CD73 (ecto 5′ nucleotidase), CD90 (Thy1), and CD105 (Endoglin) cellular markers (Fig S2A). Throughout differentiation and passages (P2-P7), MSCs markers were expressed in all hiPSC-derived MSCs (>80%) with minimal differences between conditions except for the HGPS-L line (Fig S2A). HGPS-L clones displayed a steady medium percentage of positive cells (50%) for the CD73 and CD105 markers, whereas no differences were observed for CD90 (>95% of positive cells), without inducing detectable defects in MSCs properties. This intriguing result should be explored in additional hIPSCs from HGPS-L individuals. The absence of differentiation toward the hematopoietic lineage was confirmed by the low percentage of cells positive for hematopoietic markers (e.g., <1.2% of CD34- or CD45-positive cells; Fig S2B).

At each passage, we also determined the rate of proliferation by analyzing the percentage of cells expressing the Ki67 proliferation marker in cells at 80% of confluency (Fig S2C and D). We observed a tendency toward a faster, but not significant, rate of proliferation for HGPS cells compared with young healthy donor cells with 5.14 ±1 versus 6.14 ±1 (Mean ± SD) days between passages, respectively

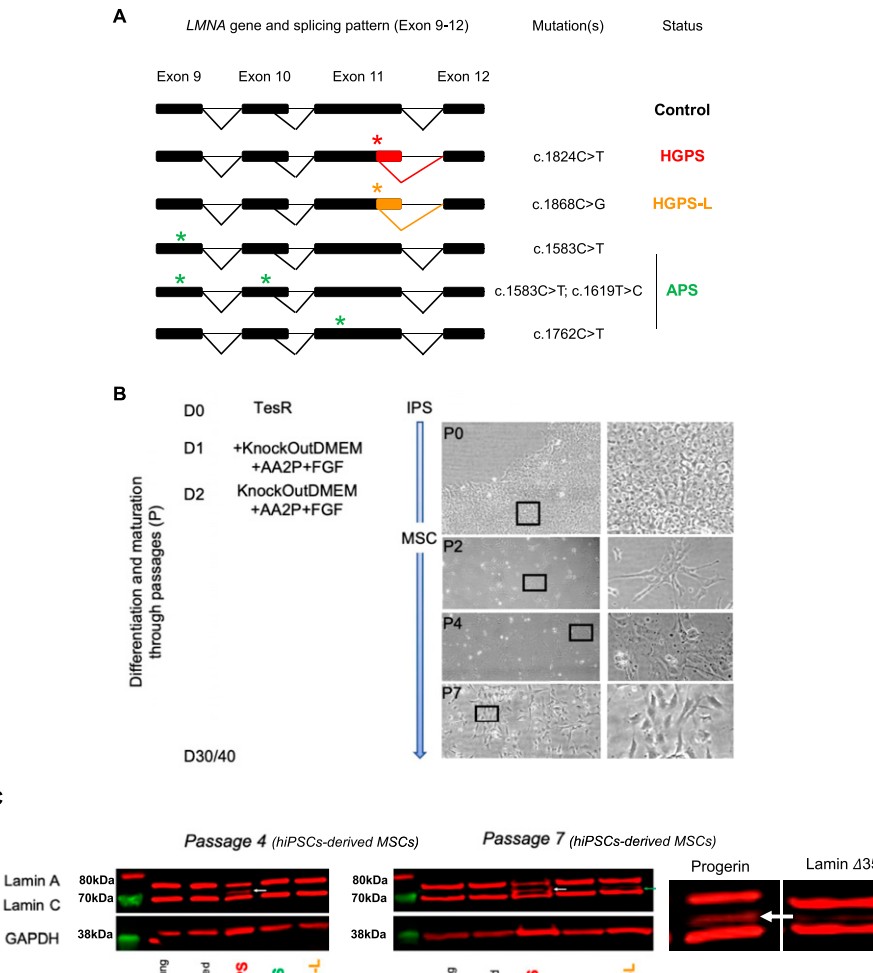

**Figure 1. hiPSCs-derived mesenchymal stem cells (MSCs) from patients with premature aging syndrome (recapitulate the cellular features observed in primary cells).**
**(A)** Schematic representation of different splicing patterns described for the *LMNA* gene and position of LMNA gene variants in Hutchinson–Gilford progeria syndrome (HGPS) (1,972, 8,243 and 5,968), HGPS-L (PC054) and atypical progeroid syndrome (OM2, 13,621, and 10,770) cells, leading to the production of pathogenic A-type lamin isoforms. **(B)** MSCs differentiation strategy. After dissociation with accutase, hiPSCs were plated on fibronectin-coated plates in the presence of ROCK inhibitor (thiazovivin). Cells are grown in TeSR, which is progressively removed upon differentiation and replaced with knockout-DMEM supplemented with ascorbic acid and FGF2. TeSR medium is removed at day 2, and cells are grown in knockout-DMEM supplemented with ascorbic acid and FGF2 until they reached 80% of confluency with medium replacement every 2 d. For the different passages, cells were dissociated with trypsin and plated in the same conditions. **(C)** Immunoblotting of A-type lamins proteins in whole-cell extracts of hiPSC-derived mesenchymal stem cells derived from young and aged healthy donors (CT-Y, CT-A); patients affected with either HGPS, atypical progeroid syndrome, or HGPS-L. MSCs were extracted at the fourth and seventh passage post-differentiation (left or right panels; respectively). Progerin is detectable in HGPS cells (Δ50 intermediate band between lamins A and C). Antibodies also detect the Δ35 isoform produced in HGPS-like cells as indicated by a white arrow and presented in the blots and their enlarged adjacent version for progerin (HGPS, passage 4) and Δ35 isoform (HGPS-L, passage 7); respectively.

(*P* = 0.79; n = 7 per condition) (Fig S2C). Next, we analyzed the production of progerin and lamins A/C at two different passages (P4 and P7) by Western blotting (Fig 1C). At P4 and P7, the level of lamin-A and -C was similar between cells, with the detection of progerin restricted to HGPS MSCs. The lamin Δ35, isoform present in HGPS-L, was detectable at P7. As expected, no progerin or truncated prelamin A could be detected in APS and Control MSCs (Fig 1C).

Hence, *LMNA* mutations and accumulation of progerin (HGPS) or truncated prelamin A (HGPS-L) did not affect the efficiency of differentiation but slightly delayed the MSCs differentiation of HGPS-L cells (Fig 1C). Accordingly, regardless of their genotype, >90% of cells expressed MSC markers at P7. For further experiments, cells were collected at the seventh passage, that is, 40 d post-differentiation for HGPS-L cells and 47 d for the healthy aged donor.

### Expression profiles differ between progeroid patients

To identify pathways dysregulated in MSCs from patients with different *LMNA* mutations, we performed RNA Sequencing in hiPSCs and hiPSC-derived MSCs (MSCs) at P7 for the different clones.

Unsupervised hierarchical clustering separated two groups, corresponding either to hiPSCs or MSCs confirming the efficient MSCs differentiation (Fig 2A). In hiPSCs, we noted only a small proportion of differentially expressed genes (DEGs) compared with controls (comparison to the young healthy donor: HGPS: 115; HGPS-L: 286; APS: 95 DEGs; comparison to the aged healthy donor: HGPS: 98; HGPS-L: 233; APS: 91 DEGs, Fig S3A). In hiPSCs, we did not identify any specifically dysregulated pathways in disease samples, even when considering DEGs that are shared between conditions. As anticipated in the absence of *LMNA* expression in pluripotent cells, this confirms that cell reprogramming erases differences between samples carrying or not *LMNA* mutations, as previously reported (22, 27, 28).

In MSCs, HGPS and HGPS-L are clustered with the aged healthy donor, whereas APS cells are clustered with the young control (Fig 2A). Compared with hiPSCs, we found a higher number of DEGs in MSCs with a 4.8-fold increase in the number of DEGs between hiPSCs and MSCs (comparison to the young healthy donor: HGPS: 452; HGPS-L: 1192; APS: 297 DEGs; comparison to the aged healthy donor: HGPS: 587; HGPS-L: 1999; APS: 324 DEGs; Fig S3B). As expected, this increase in DEGs proportions is likely stemming from the

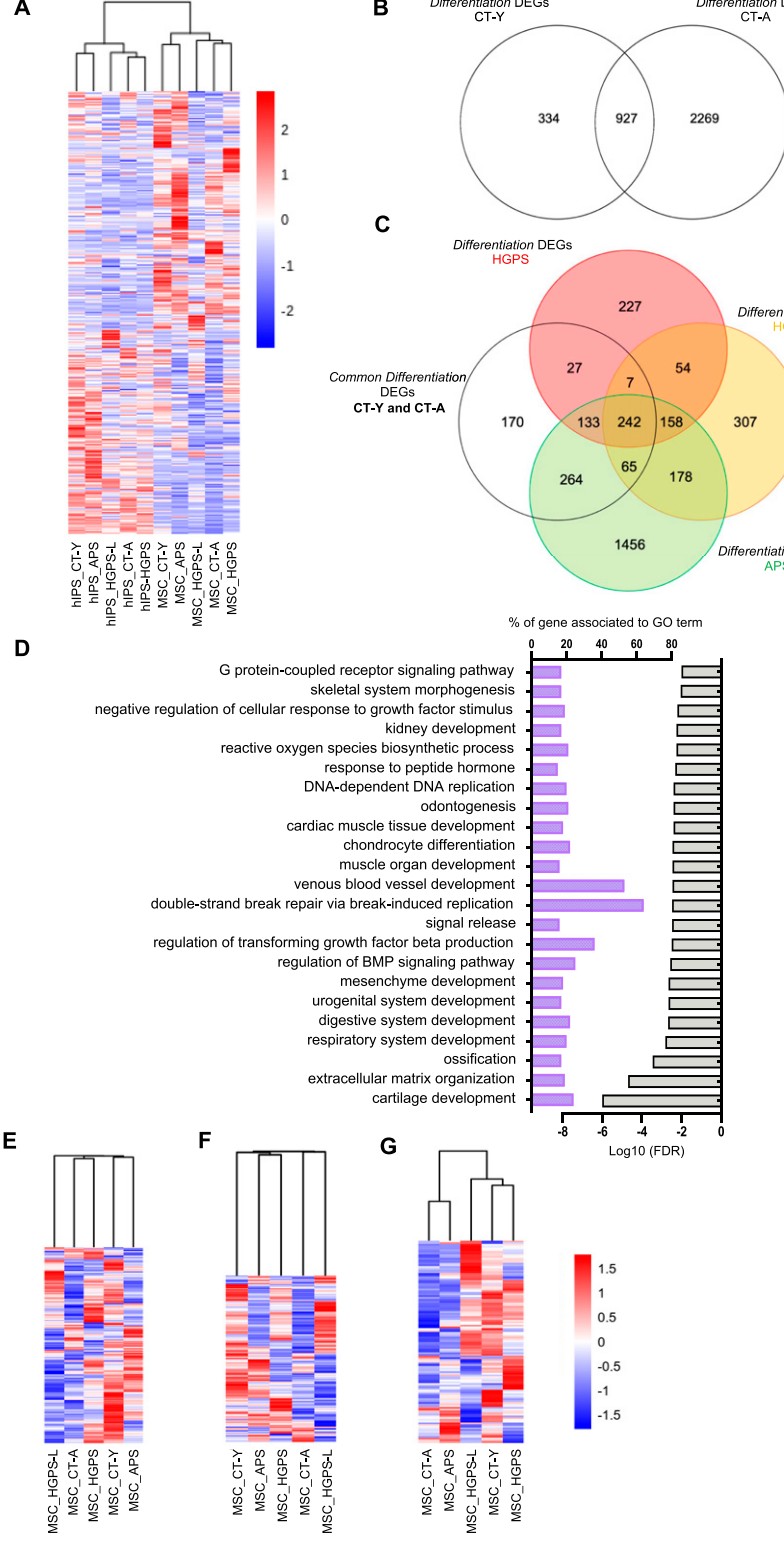

**Figure 2. Transcriptome analysis of hiPSCs and differentiated mesenchymal stem cells (MSCs) from individuals carrying *LMNA* mutations.**
**(A)** Heatmap representing the transcriptome of hiPSCs and hiPSCs-derived MSCs from progeroid patients. Analysis was performed in triplicate for most samples (CT-Y; CT-A; Hutchinson–Gilford progeria syndrome [HGPS] and atypical progeroid syndrome [APS]) and quadruplicate for HGPS-L cells. Unsupervised hierarchical clustering separates hiPSCs from MSCs. **(B)** Venn diagram between control individuals (CT-Y; CT-A). List of the differentially expressed genes during differentiation (dDEGs) were generated by comparing the transcriptome of hiPSCs to MSCs to their respective controls. We report 334 and 2269 dDEGs restricted to either control (CT-Y, CT-A; respectively) and 927 dDEGs shared between both controls. **(C)** Venn diagram of the overlap between dDEGs generated by comparing transcriptome of hiPSCs to MSCs in each pathological condition (HGPS, HGPS-L, and APS) and the dDEGs shared in controls (found in B). **(D)** Biological processes (BP) associated with dDEGs and common to all laminopathies determined using Gene Ontology (GO). dDEGs were determined as reported in respective panels (CT-Y versus pathology). Bar plots are shown with GO terms and associated percentage of genes along with the *P*-value adjusted FDR. **(E, F, G)** Unsupervised hierarchical clustering heatmap for mitochondrial (E), DNA damages (F) and telomeric (G) genes in MSCs. Analysis was performed in triplicate for most samples (CT-Y; CT-A; HGPS and APS) and quadruplicate for HGPS-L cells.

accumulation of defects caused by *LMNA* mutations during the differentiation process. At P7, comparison between conditions failed to identify any relevant pathway, probably because of the high ratio of housekeeping genes and accumulation of unspecific

defects caused by the expression of *LMNA* mutants among the DEGs.

To focus on alterations occurring at early differentiation stages, concomitant with the expression of pathogenic lamin isoforms

(progerin, lamin Δ35), we compared MSCs to their respective hiPSCs and retrieved DEGs specific to each line dubbed thereafter *differentiation* DEGs (dDEGs). For controls and independently of the donor's age, comparison between hiPSCs and MSCs retrieved 927 dDEGs (Fig 2B), thus considered as the physiological genes impacted by differentiation. We next compared these 927 dDEGs to dDEGs retrieved for each group of patients. We found 227 dDEGs in HGPS; 307 in HGPS-L, and 1,456 dDEGs in APS (Fig 2C). Using dDEGs across all pathological conditions versus controls, Gene Ontology (GO) analysis identified processes enriched for genes related to DNA replication (>60%), vascular and mesenchymal development (50% and >20%, respectively), and TGFβ regulation (40%) (Fig 2D). Interestingly, the most significant pathway in this analysis corresponds to cartilage development and organization of the ECM (20% enrichment; Fig 2D). Further GO analysis of dDEGs specific to each group of patients identified pathways related to VEGFR signaling for HGPS, several tissues/organs development in HGPS-L, or DNA replication for APS (Fig S3C).

Previous works performed in HGPS fibroblasts highlighted defects in mitochondria, DNA repair, and telomeres (9, 29, 30, 31), all associated with physiological aging (32). We thus analyzed separately DEGs that belong to these three pathways in MSCs. For genes encoding mitochondrial proteins (Fig 2E) and genes involved in DNA damage response (Fig 2F), unsupervised clustering revealed a difference between controls (young and old) and pathological samples consistent with the increase in mitochondrial network defects and DNA damage observed upon aging (32). For both biological processes, the APS line was closer to the young control as also observed when all DEGs were considered (Fig 2A). On the opposite, the analysis of DEGs corresponding to telomere-related genes (Fig 2G) revealed a clustering between APS cells and the aged control, whereas the young control was associated with HGPS and HGPS-L cells. Overall, this suggests differences in the expression profile of genes related to those three aging-associated processes between the different progeroid syndromes.

### Cells from prematurely aging patients show an overall altered DNA methylation profile

Given the links between *LMNA* gene mutations and chromatin regulation (33), we next investigated whether changes in gene expression correlate with alterations in DNA methylation in controls and cells from prematurely aged patients. To this aim, we profiled the genome-wide DNA methylation patterns of the different MSCs samples (hiPSC-derived MSCs), using the Illumina Infinium EPIC methylation assay after sodium bisulfite DNA modification. This assay that covers 850,000 CpG sites throughout the genome provides the methylation profile of the coding genome, regulatory elements, and also the noncoding genome with probes distributed in intergenic regions and gene bodies (29% and 37%, respectively), 5′ UTR (8%) and CpG located within 200 bp or 1,500 bp from transcription start sites (TSS, 12% and 8%, respectively) (34). For each probe, the methylation level of CpGs covered by the array was determined with a 99% confidence by calculating the median DNA methylation β-values and SD between samples groups (Fig 3A). Probes with an absolute mean Δ β-values of 0.2 (20%) between controls and patients were considered as differentially methylated

(DMP, differentially methylated probe) (35). Depending on the samples, we observed between 70,497 and 137,900 DMPs in pathological samples compared with controls with a global trend toward hypermethylation (Fig 3A and Table S2) that is more pronounced when comparison is made with the young donor (Fig 3B and C; CT-Y; HyperM APS, 65.5%; HGPS, 83.2%; HGPS-L, 79%) than with the aged donor (CT-A; 39% for APS, 59.15% for HGPS, and 61.43% for HGPS-L). For all comparisons, we noticed a lower proportion (15–20%) of hypermethylated probes when APS cells are compared with the young donor but a higher proportion (~20%) compared with the aged donor (Fig 3C). By comparing the percentage of probes covered by the array and categorized by CpG content using the Illumina annotations, we noticed a decreased proportion of hypermethylated probes corresponding to gene regulatory regions, that is, CpG islands, first exon TSS1500, and TSS200 (Figs 3D and E and S4A and B and Tables S3 and S4), in agreement with findings from others using different cellular models (e.g., bone marrow–derived MSCs) (36). The percentage of hypermethylated probes for open seas and intergenic regions (isolated CpG) is increased, whereas hypomethylated DMPs are localized within shores (at the border of CpG islands). Consistent with earlier observation, this trend was less pronounced in APS MSCs regardless of the control used for comparison (Tables S3 and S4).

The ChromHMM track from MSCs derived from adipocytes (E025 cells, Roadmap Epigenomics Mapping Consortium) was used to predict association to functional elements (Fig 3F and G). Regardless of the control used, methylation changes affect only slightly heterochromatin but more globally enhancer elements (Fig 3F and G and Table S5). We observed a decreased proportion of hypermethylated probes for actively transcribed regions regardless of the control used for comparison but a marked increase in the percentage of hypermethylation for quiescent genes in comparison to the young donor (Table S4). This feature might be associated with aging, as reported by a previous study (36), because the repartition of CpGs associated to quiescent genes is similar between patients and the aged donor (about 18% of hyper- and hypomethylated DMPs, Table S5).

### Methylation changes mainly concern lamina-associated domains in patients with premature aging

Reports in HGPS fibroblasts suggest that genes dysregulated in premature aging syndromes are mostly enriched in lamin-associated domains (LADs) (37, 38). To whether this is the case in the different disease contexts, we specifically analyzed DEGs and DMPs that correspond to LADs. Comparison of genes covered by the RNA-Seq analysis (n = 60,733 from GENCODE annotation) to the list of DEGs (from MSCs) in each condition and their respective distances to LADs did not reveal any massive enrichment of genes located close to LADs (median distance in all conditions = 1.7 Mb; Fig S5A). However, plotting DMPs to their respective distance from LADs identified two groups. The first group comprises HGPS, HGPS-L, and the aged donor cells (CT-A); the second group, the APS together with the young donor cells (APS, CT-Y; Figs 4 and S4C), as for the transcriptome analysis (Fig 2A). Within LADs, the group corresponding to HGPS, HGPS-L, and the aged donor displayed a high and narrow peak for highly methylated probes, visible as a fixed

**MSCs-hIPSCs derived**

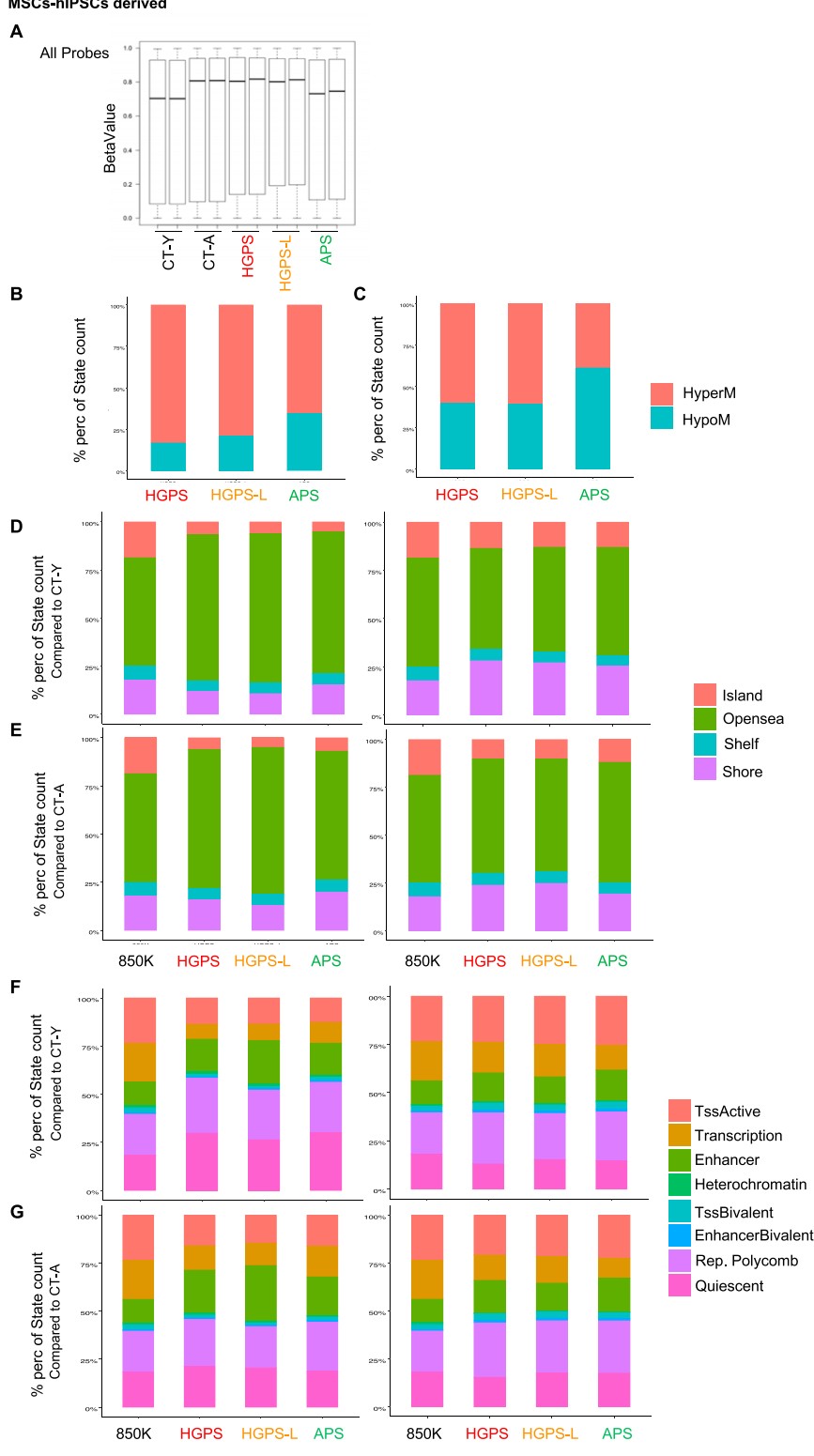

**Figure 3.    Analysis of global methylation in hiPSC-derived mesenchymal stem cell (MSC).**
**(A)** Median of methylation Beta values for all CpG probes in the different conditions. Median is represented by the black bar. **(B, C)** Stacked barplots representing the distribution of differentially methylated probes in patient cells (MSCs) compared with healthy young (B) and aged (C) donor cells ($P$adj < 0.05 and abs($\Delta\beta$) > 0.2). **(D)** Stacked barplots representing the distribution of hypermethylated (left) and hypomethylated (right) probes relative to CpG islands, shores (2 kb flanking CpG islands), shelves (2 kb extending from shores), or open seas (isolated CpG in the rest of the genome) in patients compared with young donor cells (CT-Y; $P$adj < 0.05 and abs($\Delta\beta$) > 0.2). **(E)** Stacked barplots representing the distribution of hypermethylated (left) and hypomethylated (right) probes relative to CpG islands, shores (2 kb flanking CpG islands), shelves (2 kb extending from shores), or open seas (isolated CpG in the rest of the genome) in patients compared with aged donor cells (CT-A; $P$adj < 0.05 and abs($\Delta\beta$) > 0.2). **(F)** Stacked barplots representing the distribution of hypermethylated (left) and hypomethylated (right) probes analyzed for REMC (Roadmap Epigenomics Mapping Consortium) features for adipose-derived MSC cultured cells (E025), in patients compared with young donor cells (CT-Y). Features with similar characteristics were pooled for better visualization on the plot. ($P$-value < 0.05 and abs($\Delta\beta$) > 0.2). **(G)** Stacked barplots representing the distribution of hypermethylated (left) and hypomethylated (right) probes analyzed for REMC (Roadmap Epigenomics Mapping Consortium) features for adipose-derived MSC cultured cells (E025), in patients compared with aged donor cells (CT-A). Features with similar characteristics were pooled for better visualization on the plot. ($P$-value < 0.05 and abs($\Delta\beta$) > 0.2). In all stacked barplots presented (D, E, F, G), we report the distribution given by the whole EPIC Array (850k probes, Illumina).

yellow region associated with a superior median value of methylation at LADs (Fig 4B, CT-A); whereas the second group (APS, CT-Y) exhibited a lower and more spread methylation profile, illustrated by two yellow regions and a lower median value at LADs when compared with others (Fig 4A, CT-Y; Welch two-sample t-test $P < 2.2 \times 10^{-16}$ for all pathologies compared with controls). We further confirmed these observations by comparing group effect, as defined by Cohen's test (i.e., d < 0.2: no effect), across conditions, CT-Y

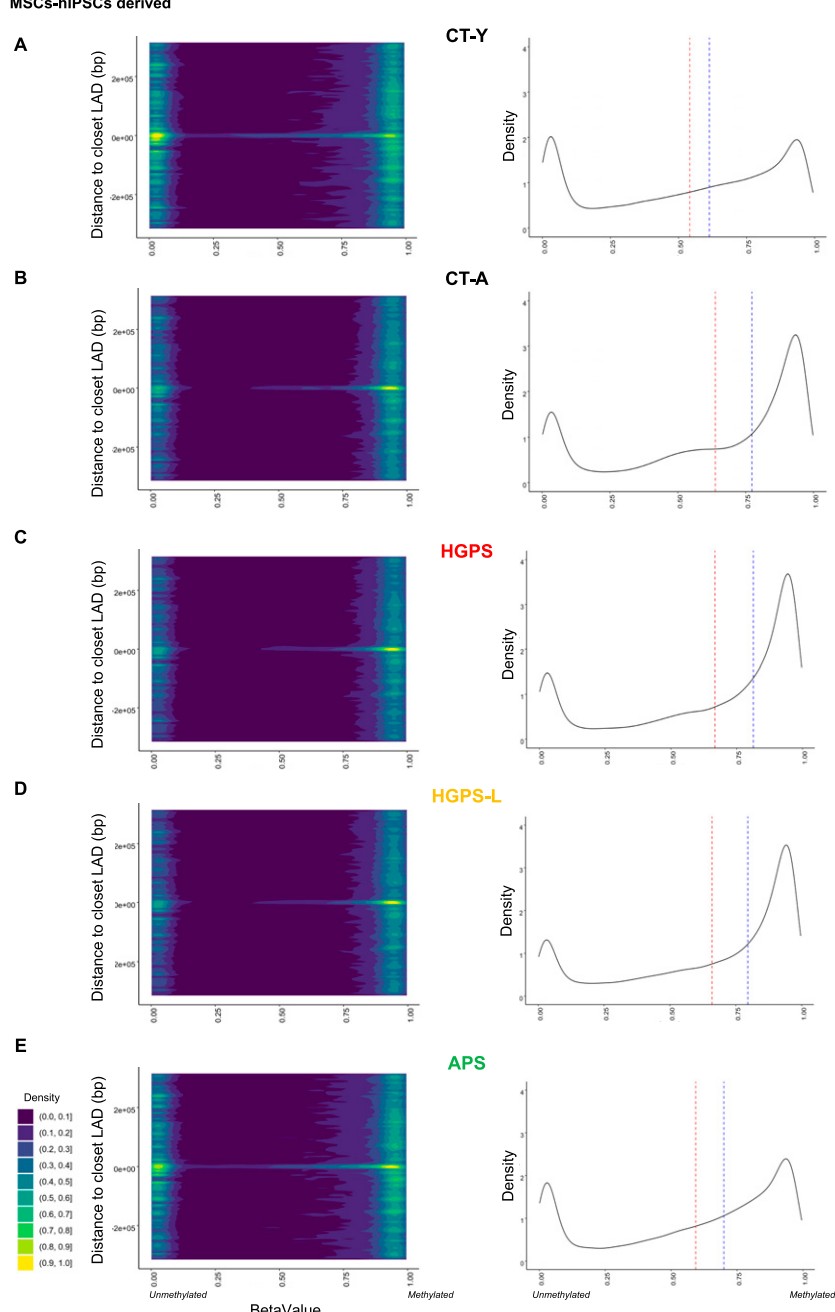

MSCs-hIPSCs derived

**Figure 4. DNA methylation regarding lamin-associated domains in hiPSC-derived mesenchymal stem cells.**
2D density plots of distances to closest LAD and associated probe methylation (left panels) and density plot of betavalue (methylation) of probes located within LADs (right panels). **(A, B, C, D, E)** We report results from the young control (CT-Y, A); aged control (CT-A, B), progeria (Hutchinson–Gilford progeria syndrome, C), progeria-like (Hutchinson–Gilford progeria syndrome-L, D), and atypical progeroid (atypical progeroid syndrome, E) samples; respectively. For 2D density plots, we report all probes located within ±315 kb of LADs (n = 167,848 probes) that successfully passed the QC threshold (GSM1313397). Probes located within LADs are associated to 0. Probe density is represented by a color scale where yellow represents the highest density. Methylation levels are reported by their associated betavalue where 0 report unmethylated probes; 1 fully methylated. For density plot located in LADs (n = 81,842 probes), dashed red line indicates mean methylation; dashed blue line indicates median methylation. We report a higher density of unmethylated probes at LADs in CT-Y and atypical progeroid syndrome hiPSC-derived mesenchymal stem cells as seen by a yellow signal at 0 on both axis in their 2D density plot panels (left panels) and further confirmed by a lower median axis (blue dashed lines) in the density plot at LAD (right panels).

versus HGPS (d = 0.4), CT-Y versus HGPS-L (d = 0.4), CT-Y versus APS (d = 0.2). Of note, because of an overrepresentation of CpGs within LADs (n = 81,842, 10% of total), we performed a similar analysis, keeping only CpGs outside of LADs. Again, we note an identical distribution of groups with immediate LAD shores mostly hypomethylated in APS and CT-Y samples compared with HGPS, HGPS-L, and CT-A (Fig S4C).

Overall, our comparison of MSCs from patients with premature aging, and accumulating or not progerin or truncated prelamin A, suggests that in these cells, *LMNA* splice mutations (HGPS and HGPS-L) or point mutations (APS) have different consequences on

the epigenetic and transcriptomic landscape with HGPS and HGPS-L cells more closely resembling to cells from an aged donor than APS cells.

## Selective gene deregulation through differentiation in MSCs

To further delineate the impact of *LMNA* mutations in MSCs, we selected a subset of genes involved in pathways related to MSC differentiation (Figs 2D and 3C; n = 8) and performed RT-qPCR at P7. We analyzed expression of *GTF2H2* (DNA repair); *FOXC1* and *FOXC2* (mesenchyme development); *COL1A1* and *COL1A2* (cartilage

development); *TBX20* (heart development); and *TGFB2* or *FGFR2* (growth signaling cascade) in the samples analyzed by RNA-Seq and additional ones (HGPS, n = 3; HGPS-L, n = 1; APS, n = 3). We confirmed significant differences in gene expression between MSCs from healthy donors and patients (Fig S5B). Down-regulation of *FOXC1* is restricted to APS cells ($P < 0.05$; Kruskal–Wallis test). *COL1A1* and *COL1A2* are down-regulated in cells producing progerin or truncated prelamin A (HGPS; HGPS-L, $P < 0.01$; Kruskal–Wallis test) but up-regulated in APS cells ($P < 0.01$; Kruskal–Wallis test). Next, we analyzed the kinetics of expression of these genes at different passages to ask whether gene modulations in patients (i.e., linked to progerin or truncated prelamin A accumulation) might be a consequence of the differentiation process. Strikingly, in APS cells, *FOXC1* expression increases transiently at P4, whereas *FOXC1* expression seems to increase gradually in all other conditions. Likewise, *COL1A1* and *COL1A2* expression increases between passages, with the exception of *COL1A2* in APS cells that displayed a steady expression (Fig S5C). These observations suggest that defects observed at P7 could be the consequences of early events already in place at P2 (HGPS; HGPS-L) and persisting throughout differentiation. However, in APS cells, changes in gene expression might be more transient (*FOXC1*, P4). For some candidates (*FGFR2*, *TBX20*, and *TGFB2*), the trend of modulation (up- or down-regulation) is not consistent between samples, suggesting variation in the differentiation process.

One could hypothesize that the accumulation of nuclear abnormalities along with accumulation of progerin or truncated prelamin A progressively impacts chromatin organization and gene expression. This would result in a stochastic expression of genes in cell populations from each independent differentiation replicate, in agreement with our observations on transcriptomic data (39, 40).

### Premature aging linked to the *LMNA* gene mutations leads to increased nuclear anomalies and DNA damage response, with different impacts on telomere erosion

Our previous analyses revealed an unpredicted clustering between pathologies with APS separated from HGPS and HGPS-L and confirmed that gene expression, chromatin structure, and their respective links to LADs profile are impacted by the consequences of mutations in the *LMNA* gene (Figs 2 and 4). We thus further investigated the main pathways evidenced in all MSCs (e.g., control, n = 2; HGPS, n = 3; HGPS-L, n = 1; APS, n = 3). As nuclear abnormalities are a hallmark of HGPS primary cells (41), we first evaluated the percentage of MSCs presenting nuclear deformations in the different samples at P7. This passage corresponds to the highest percentage of MSCs in each population and the presence of progerin in HGPS and lamin Δ35 isoform in HGPS-L cells. On average, the young control displayed 8% (±SD = 1) of cells with nuclear abnormalities (Fig 5A and B). This rate increases significantly ($P < 0.0001$; Tukey's multiple comparisons test) in the aged control (Mean ± SD: 27% ± 16), HGPS (45% ± 26), and APS (56% ± 21) and reached a maximum of 72% (±SD = 3) for HGPS-L cells.

In a second step, we quantified DNA double-strand breaks (DSB) by performing γH2AX staining. Concomitant with the expression of progerin and accumulation of nuclear abnormalities (Fig 5A), we observed a significant increase in γH2AX foci in HGPS, HGPS-L, and APS cells compared with controls ($P < 0.0001$, Holm-Sidak's multiple

comparisons test; Fig 5C and D). By considering more specifically the accumulation of γH2AX foci at telomeres, we found a marked and significant increase in telomere induced foci (TIFs) in the different diseases compared with controls ($P < 0.01$ for HGPS; $P < 0.0001$ for HGPS-L and APS; Holm-Sidak's multiple comparisons test) that correlates with the loss of telomeric signal in HGPS but not in HGPS-L and APS (Fig 5D).

Overall, these results indicate that despite the suppression of age-related markers (i.e., DNA damage, nuclear defects) by reprogramming, production of mutated A-type lamins during MSC differentiation recapitulates the cellular defects observed in primary patient's cells with an even higher rate of age-associated defects in HGPS-L and APS cells compared with HGPS.

### MSCs from patients with premature aging display an altered mitochondrial pattern

Increased DNA damage response and repair (DDR) and accumulation of reactive oxygen species (ROS) associated with altered mitochondrial pattern has been reported in HGPS cells (42, 43, 44, 45) but never investigated in APS and HGPS-L. To further address whether cellular stress and DNA damage correlates with mitochondrial defects, we analyzed the mitochondrial network of MSCs by confocal microscopy after staining with mitotracker. After 3D image reconstruction using the Imaris software (Bitplane), we counted cells with a normal mitochondrial pattern (Fig 5E and F) and cells in which this pattern is modified, that is, segmented, reduced, or clustered around the nucleus. In young control MSCs, more than 90% of cells (Mean ± SD: 97% ±4) displayed well-organized mitochondrial networks around the nucleus with spreading throughout the cytoplasm. In agreement with earlier report on mitochondrial aging (46, 47), the number of cells with a fragmented mitochondrial network increases significantly in MSCs from the aged healthy donor (51% ± 54, $P < 0.0001$; Tukey's multiple comparisons test). In cells from patients with premature aging (HGPS, HGPS-L and APS), the mitochondrial network presented distinctive patterns and appeared more fragmented and concentrated at the periphery of the nucleus in APS, HGPS, and more particularly in HGPS-L cells (87% ± 16, $P < 0.0001$; Tukey's multiple comparisons test). Consistent with previous findings (9), our data further show that mitochondrial defects occur at early stages and are present in precursor MSCs with possible repercussion in final differentiation toward the different MSC-derived lineages (i.e., adipocytes and osteocytes).

Altogether, our data showed common hallmarks in MSCs derived from patients with premature aging syndromes linked to *LMNA* mutations (DNA damage response and nuclear and mitochondrial abnormalities), regardless of the type of mutation, but possibly linked to different pathways depending on the presence (HGPS, HGPS-L) or absence (APS) of truncated prelamin A isoforms, including progerin.

## Discussion

For the past decade, hiPSCs have allowed to explore the etiology of a wide range of diseases, thanks to the direct in vitro modeling of

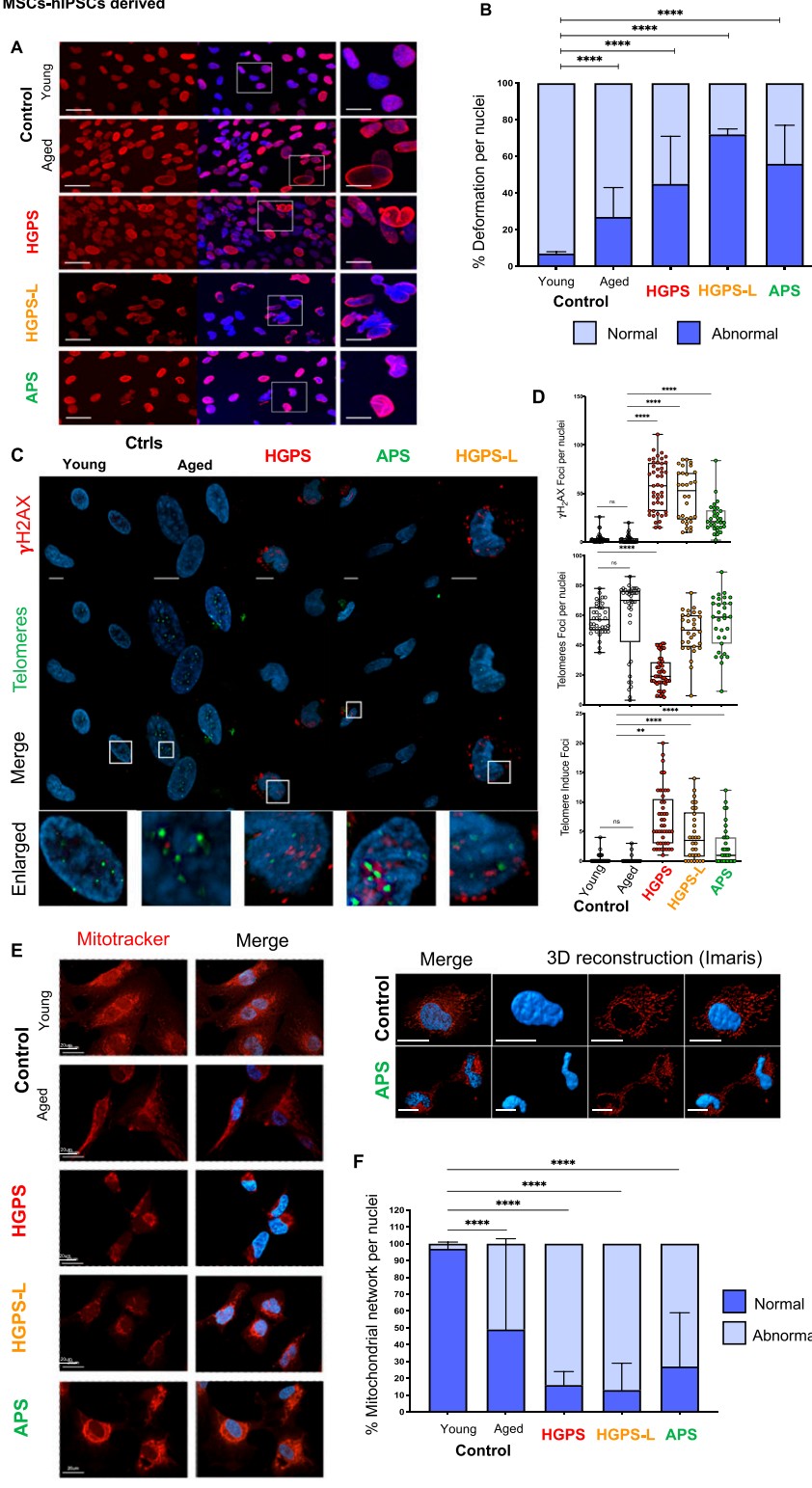

**Figure 5. hiPSC-derived mesenchymal stem cells (MSCs) from patients with premature aging syndromes display hallmarks of aging.**
**(A)** Representative Z-stack after confocal microscopy showing the distribution of lamins A/C (red) in the nucleus of hiPSC-derived MSCs (controls, Hutchinson–Gilford progeria syndrome [HGPS], HGPS-L, and atypical progeroid syndrome) at passage 7. Cells were immunostained using anti-lamin A/C antibodies and were counterstained with DAPI. Bars correspond to 30 $\mu$m (left panels) and 10 $\mu$m (right panel, enlargement) scales; respectively. **(B)** Quantification of nuclear abnormalities in hiPSC-derived MSCs. Experiments were carried in triplicates; 100 nuclei were counted per condition. The percentage of nuclear abnormalities observed was plotted. Mean ± SD are reported. Tukey's multiple comparisons test *P*-value * < 0.05; ** < 0.005; *** < 0.0005; **** < 0.0001. **(C)** Immunostaining of γH2AX foci in the different hiPSC-derived MSCs at P7 (upper panel). Telomeres were visualized using a telomeric PNA FISH probe. Telomere-induced foci (TIFs) were evidenced by overlays between γH2AX staining and telomeric signals. **(D)** For each assay, we report the number of telomeric foci, γH2AX foci along with TIFs. Experiments were carried out in duplicate with a minimum of 50 nuclei counted by condition. Mean ± SEM are reported. Significant differences are shown using the Holm-Sidak's multiple comparisons test *P*-value * < 0.05; ** < 0.005; *** < 0.0005; **** < 0.0001. **(E)** Mitochondrial network in MSCs. Staining was performed using mitotracker and counterstained with DAPI. **(F)** Quantification of cells with normal or abnormal mitochondrial patterns. Experiments were carried out either in triplicate (CT-Y, CT-A, and HGPS-L) with 100 nuclei counted per replicate (n > 300 nuclei per condition) or across biological triplicate (HGPS, atypical progeroid syndrome) with 100 nuclei counted per individual (n > 300). We report the percentage of nuclear abnormalities observed by immunofluorescence at P7; mean ± SD. Significant differences are shown using Tukey's multiple comparisons test *P*-value * < 0.05; ** < 0.005; *** < 0.0005; **** < 0.0001.

affected tissues. Nevertheless, to date, most of the information on the pathophysiology of HGPS and related premature aging syndromes comes from studies performed on patient-derived skin fibroblasts or animal models (15, 18, 29, 31, 38, 41). Besides, other

mutations of the *LMNA* gene have been found in patients with characteristics of premature aging (2, 3, 15, 16, 17, 18, 19, 20, 21), but very little is known on the molecular consequences of lamin A alterations. A common feature of patients affected with lamin-

associated mutations is the premature aging of MSC-derived tissues (e.g., myocardium, cartilage, bones, skin, and vascular smooth muscle cells). However, the mechanisms linking the production of altered lamin A to MSC differentiation defects remain unknown. Here, using reprogrammed cells from patients affected with HGPS, progeria-like syndrome (HGPS-L) or APS, we have broadened the description of the phenotypic cellular alterations linked to causative *LMNA* mutations. Overall, we describe that the accumulation of defects associated with cellular aging is common to all samples, whereas DNA methylation profiles differ between HGPS or HGPS-L and APS MSCs.

To further identify pathways associated with MSC defects which are specific to premature aging, we performed a transcriptome analysis of the different pathological samples in hiPSCs and derived MSCs. Importantly, we set our exploration in hiPSCs with low passages as the persistent maintenance of hiPSCs might induce a drift unrelated to their associated mutations ([48], [49]). Likewise, all MSCs were analyzed at P7, the earliest passage where progerin and isoform (lamin Δ35) could robustly and consistently be detected. Of note, because cells were kept in culture until P12, and no differences in proliferations were observed at P7, we considered that P7 represent the best timing to investigate the impact of *LMNA* mutations with minimum bias induced by the overaccumulation of defects on cellular senescence and proliferation.

Clustering analysis of RNA sequencing data revealed massive changes in gene expression between hiPSCs and MSCs, consistent with the absence of A-type lamins expression in pluripotent stem cells and induction of gene expression upon differentiation. The low number of DEGs in control hiPSCs compared with hiPSCs from premature aging patients indicates that despite the presence of nuclear abnormalities and progerin accumulation in the primary fibroblasts, reprogramming to the pluripotent state erases most of the differences as previously reported ([27]), resulting in a very similar gene expression profile between patients and healthy donors.

As in hiPSCs, the lists of DEGs retrieved from hiPSC-derived MSCs did not discriminate biological pathways or samples with *LMNA* mutations from one another. An observation probably because of the cumulative effects of increased expression of *LMNA* and associated mutations (progerin and lamin Δ35) during early and late differentiation into MSCs. We thus focused on DEG retrieved from the differentiation process (dDEGs), to narrow down our analyses to the earliest (and most specific) events. Taking advantage of our transcriptome in both hIPSCs and MSCs, we generated a list of dDEGs (hIPSC versus MSC) for each sample corresponding to gene differentially expressed specifically during differentiation. Next, to interpret dDEGs in the context of diseases, we subtracted dDEGs retrieved from both controls (CT-Y; CT-A), considered as representative of the physiological differentiation process. We then obtained dDEGs respective of each *LMNA*-associated mutation, modeling the potential early lamin-associated defects. Interestingly, most dDEGs correspond to biological pathways related to the development of the mesoderm, cartilage, ECM, and response to transcription factors, consistent with defects reported on these pathologies ([12], [16], [17], [20], [21]). Among dDEGs, we selected a list of genes corresponding to the identified biological pathways,

including two collagens (*COL1A1*, *COL1A2*) and one transcription factor (TF) involved in the development of mesenchymal tissues, bone, skin, and the cardiovascular system, *FOXC1* ([50],[51]). We confirmed the constant down-regulation of these ECM proteins, specifically in cells producing progerin or the Δ35 isoform (HGPS and HGPS-L, respectively). Hence, a small proportion of cells producing progerin are delayed in the production of ECM proteins (COL1A1 and COL1A2). At P7, where we could detect robust progerin and isoforms production, *COL1A1* and *COL1A2* levels are still decreased, suggesting that no efficient compensation occurred. In our hands, *FOXC1* down-regulation is restricted to APS MSCs, with a transient up-regulation at P4, concomitant with the highest replication rate observed in these cells (P3 to P4). *FOXC1* is a short-lived TF ([51]), selectively expressed in chondrocytes ([52]), and involved in hair maintenance ([53]). Changes in gene expression suggest a disturbed process in the orchestration of TFs required for MSCs differentiation. Regulation of ECM proteins in HGPS, HGPS-L, and TF in APS remains to be investigated, including in more defined cell types as all MSCs were capable of reaching final differentiations stages and produce chondrocytes, osteoblasts, or adipocytes.

During aging, the overall level of methylation tends to decrease over time with a local hypermethylation of specific sequences ([54], [55]). Yet, this trend might depend on the type of cells as the opposite was reported in muscle stem cells upon aging ([56]). To globally evaluate the impact of *LMNA* mutations on the epigenome, we analyzed DNA methylation profiles in the different samples. Our study highlights hypermethylation of HGPS and HGPS-L patients with a similar distribution of methylated CpG probes as in the aged control cells. In APS cells, the distribution of hyper- or hypomethylated probes resemble the young control. Although reprogramming can reset the epigenetic aging clock, it has been demonstrated that a significant residual signature of the donor age remains in hiPSCs ([57]), whereas most can be erased by extensive hiPSCs expansion along with an increased risk of genetic alteration and epimutations ([58]). An overall resistance to demethylation has been observed in hiPSCs from elderly donors; in accordance with this, age-related CpGs in hiPSCs derived from older donors display increased methylation levels compared with hiPSC derived from young individuals. Because we kept our hiPSCs at low passages, this residual age-associated signature is most likely carried on in our hiPSCs, conserved upon differentiation into hiPSC-derived MSCs and explains the differences in methylation profiles observed between our young and old donors control cell lines.

Regarding probe distribution, we observed an increase in hypermethylated probes in intergenic regions and a decrease of hypermethylated probes at transcription initiation sites, consistent with previous report ([36]), an impact of promoter methylation on gene expression ([59]) and an overlap between down-regulated DEGs and aggregated hypermethylated DMPs. Accordingly, we found the highest overlap in methylation and transcription when considering down-regulated DEGs and aggregated hypermethylated DMPs (Table S6).

In all samples associated to *LMNA* mutations, we observed an altered distribution of the mitochondrial network, increased chromosomal abnormalities, and an accumulation of DNA damage

response foci associated or not with telomere erosion; all corresponding to obvious signs of aging (32). Premature aging of HGPS patients and accumulation of progerin caused by the *LMNA* c.1824C>T gene mutation summarize most of these cellular markers (13, 60). Increase in telomere induced foci (TIF) correlates with the loss of telomere signals (61, 62). Consistently, by comparing telomeres in HGPS-derived MSCs to an age-matched control and cells derived from a healthy 82-yr-old donor (CT-A), we observed that the drastic decrease in the number of telomere signals in HGPS cells associates with accumulation of double-strand DNA breaks as a response to DNA damage. Telomere erosion is not as marked in HGPS-L cells and APS cells. In these cells, the *LMNA* mutations are paralleled with accumulation of γH2AX foci at telomeres, suggesting telomere attrition. However, in these cells (HGPS-L, APS), the number of telomere signals does not decrease significantly (e.g., no significant loss of telomere ends), whereas the number of TIFs is increased. This might further indicate either a slower rate of telomere erosion, a high number of cells with critically short telomeres but a lower number of cells with telomere loss (i.e., no signal) or telomere uncapping. Of note, we did not evidence any change in the expression level of genes encoding proteins forming the shelterin complex (Fig 2G), including TRF2 that interacts with A-type lamins but not with progerin (63), in HGPS, HGPS-L, APS contexts. Intriguingly, compared with HGPS and HGPS-L cells, we did not observe any change in APS MSCs replication rate, accumulation of nuclear abnormalities, or DNA damage foci except for TIFs. Thus, this suggests converging pathways that seem unrelated to DNA damage and senescence as replication rates remains similar, even long after initiation of differentiation followed by the maintenance of cells in culture (P12) but potentially linked to TFs and/or methylation patterns, as highlighted above.

The main cause of DNA damage and telomere erosion, apart from replication, is the accumulation of ROS. In the literature, it is widely accepted that the primary source of ROS is the mitochondria, as a normal by-product of the respiratory chain (64). We observed a modification of the mitochondrial network with a condensed and concentrated mitochondrial network distribution in the periphery of the nucleus in patients, whereas controls showed a more diffuse distribution across the cytoplasm. In addition, our RNA-Seq analyses showed that several genes encoding mitochondrial proteins are differentially expressed in patient cells and the elderly control when compared with the young control. Importantly, all our observations (nuclear shapes, DNA damage, and mitochondrial network) are made under basal conditions. Further studies are needed to fully grasp changes detected across pathologies. This might include the use of stressor and protector of DNA damage or ROS and antioxidant (mitochondrial network). Depending on their associate *LMNA* mutations, differences observed could be erased (because of resilience) or exacerbated.

Overall, we showed that mutations in *LMNA* leading to premature aging, associated or not with progerin or other truncated prelamin A isoforms accumulation, impact processes related to DNA structure (methylation and DNA damage), nuclear shape, and mitochondria. All of them are hallmark features of aging and occur at early stages of differentiation of MSCs. In addition, beyond progerin clearance,

our work suggests that ECM defects supporting differentiation could be at play in HGPS, whereas transcription factor seems transiently affected in APS. Furthermore, our work describes common aberrant features between pathologies and hiPSC-derived MSCs from an old donor with the exclusion of DNA methylation. Indeed, hiPSC-derived MSCs from APS patients and the young donor share similar DNA methylation pattern, suggesting that DNA methylation might not be a key factor in premature aging. This observation is complementary to previous work (22, 36) but requires a higher number of samples (we report 7 *LMNA*-associated pathologies and two controls) before additional speculations, because of potential compensations in APS cells for example.

As recently performed in mice (10, 65), future studies using our cellular model could assess current leading treatments for HGPS (e.g., farnesyl transferase inhibitor and rapamycin analog to promote clearance) and their potential effects in associated *LMNA* diseases (HGPS-L), including in the absence of progerin accumulation (APS). Taking advantage of hiPSCs and their derived MSCs, treatment could be performed as early as possible during the differentiation process to appreciate if one can reverse modestly (or completely) aging hallmarks (nuclear abnormalities, DNA damage) as recently suggested (66) and reported in in vitro (14) and in vivo models (10, 65, 67).

Although HGPS and HGPS-L cells may display the cumulative impact of progerin (or other toxic isoforms), APS cells, which do not produce progerin, exhibit an early disturbed differentiation process that ultimately leads to the same cellular aberrations, opening new grounds for understanding tissue-specific defects in premature but also physiological aging.

# Materials and Methods

### Human ethics statement

Parents have provided written informed consent for the use of biopsies for medical research in accordance with the Declaration of Helsinki. Samples were provided by the Center for biological Resources (Department of Medical Genetics, La Timone Children's Hospital) with the AC 2011-1312 and N°IE-2013-710 accreditation numbers (Table S7).

### Induced pluripotent stem cells (iPSC) generation

Isolated primary fibroblasts were reprogrammed into iPS cells as reported elsewhere (24, 68) using the OKSM cocktail. For each reprogramming, at least 10 clones were collected. Two clones were fully validated for each sample for alkaline phosphatase activity, expression of pluripotency markers (flow cytometry and RT-qPCR), chromosomal stability (karyotyping), and the capacity for each clone to differentiate and generate embryoid bodies expressing markers of the three embryonic lineages. All iPSC cells used in this study fulfilled these criteria. Colonies were grown and expanded in mTeSR1 medium (Stemcells) on BD Matrigel (BD Biosciences, Cat. no. 354277) coated dishes.

## MSC differentiation

HiPSC-derived MSCs were grown on poly-D-lysine (1 mg/ml)–pretreated culture dishes coated with fibronectin (0.1%; Sigma-Aldrich). For differentiation, IPS cells were plated in mTesR medium at day 1. At day 2, the medium was switched to a mixture composed of TesR and Knock-Out DMEM (KO-DMEM) basal media (1:1). KO-DMEM was supplemented by FBS (15%; Life Technologies), GlutaMAX (1×; Life Technologies), nonessential amino acids (1×; Life Technologies), antibiotics (penicillin, streptomycin), and β-mercaptoethanol (50 μM). Medium was supplemented with thiazovivin (5 μM; StemCell), β-FGF (10 ng/ml; Peprotech), and L-ascorbic acid-2-phosphate (1 mM; Sigma-Aldrich). At day 3, cells were switched to a KO-DMEM medium supplemented with 15% FBS, thiazovivin (5 μM; StemCell), basic FGF (10 ng/ml; Peprotech), and L-ascorbic acid-2-phosphate (1 mM; Sigma-Aldrich). Cells were split at 90% confluency with accutase (StemCell).

## Flow cytometry

Cells were treated with accutase (StemCells) and counted with a KOVA glass slide (Thermo Fisher Scientific). Cells were then spun at 500$g$ for 5 min and resuspended in a buffer composed of PBS (1×) supplemented with FBS (2.5%; Life Technologies) and sodium azide (0.1%) at 1milion cells per ml. We used 100 μl of cells (100,000 cells) per condition. Antibodies were added to each tube and incubated for 2 h at RT (protected from light) then washed thoroughly with FACS buffer for a total of 3 × 5 min washes.

For hiPSCs characterization, SSEA4, Tra-1-60, and TRA-1-81 antibodies were used. For MSCs, we used CD105 (Endoglin, 12-1057-42; Thermo Fisher Scientific), CD90 (Thy1, 559869; BD Biosciences), and CD73 (ecto-5′-nucleotidase, 550257; BD Bioscience) as positive markers and CD45 (563204; BD Biosciences) with CD34 (561440BD; Biosciences) antibodies as negative hematopoietic lineage marker. We used an isotype control for each condition corresponding to the fluorochrome of the antibody. Cells positively labeled for both CD73 and CD105 cells were considered as MSCs. Subsequent experiments in this study using MSCs were only performed using populations of cells that reached >80% of positive (CD73[+]; CD105+) cells per condition.

## Immunofluorescence assay

Immunofluorescence assays were performed as followed: cells were grown on cover slides and fixed for 10 min with 4% para-formaldehyde diluted in 1× PBS. After PBS washes (3 × 5 min), cells were incubated for 15 min at RT in a permeabilization solution (0.5% Triton X-100 in PBS) before incubation for 2 h at 37°C in a blocking solution (2 mg/ml BSA in PBS). Cells were then incubated overnight at 4°C in a blocking solution containing the primary antibodies (anti-lamin A/C; 1:100, Sc6215; Santa Cruz, anti-γH2AX 1:250, ab11174; Abcam). After three washes with PBS/0.1% Tween (3 × 5 min), slides were incubated for 45 min at RT with the secondary antibody (Alexa Fluor 1:500) in PBS 1× supplemented with BSA (1 mg/ml). Slides were mounted in VECTASHIELD with DAPI (Vector Laboratories), and images acquired using a confocal imaging system (LSM800; Zeiss).

## TIFs

Cells were grown on cover slides and fixed for 10 min on ice with 4% paraformaldehyde in 1× PBS. After PBS washes (3 × 5 min), cells were incubated for 1 h at RT in blocking solution (1% Triton X-100, 1% BSA, 5% donkey serum in PBS). To perform the PNA-FISH staining, cells were washed twice with SSC2X for 5 min at RT and subsequently treated with RNase A for 45 min at 37°C. After an additional SSC2X wash (5 min, 4°C), cover slides were dried and incubated upside-down with a hybridization solution containing the PNA probe (20 μl H2O; 70 μl formamide; 7 μl 10% blocking B [Roche]; 1 μl 1M Tris, pH 7.2; 1 μl probe) and sealed on coverslips using rubber cement. Slides were then heated at 85°C for 4 min and incubated in the dark at 37°C in a humidification chamber for 2 h. After removal of the rubber cement, cells were serially washed in three different solutions: twice for 15 min at RT with washing solution I (10 mM Tris, pH 7.2; 70% formamide); twice for 15 min at RT with washing solution II (150 mM NaCl; 50 mM Tris, pH 7.2; 0.05% Tween 20); and twice with 1× PBS for 5 min at RT. Cells were then blocked for 1 h with the blocking solution and immunostained overnight at 4°C in the blocking solution containing the primary rabbit polyclonal anti-53BP1 antibody (1:500; Novus Biologicals). After three washes with PBS/0.1% Triton X-100, slides were incubated for 1 h 30 min at RT with Alexa 555 Donkey anti-rabbit secondary antibody in PBS containing 0.5% Triton X-100, 1% BSA, 2.5% donkey serum. Slides were mounted in VECTASHIELD with DAPI (Vector Laboratories). Images were taken using a confocal system (LSM800; Zeiss). Co-localization events, representing telomeric DNA damages (TIFs), were counted in at least 30 nuclei per condition from three independent experiments using the IMARIS software.

## Western blot

Cells were collected by centrifugation and stored at 1 million cells per aliquot at −80°C for future use. Cell lysis was carried out using a lysis buffer composed of Tris–HCl, pH 7 (100 mM), EDTA (10 mM), glycerol (10%), and SDS (10%). Samples were then frozen at −80°C and thawed for a total of three cycles and finally stored at −80°C. Before loading, samples were diluted in migration buffer composed of cell lysate (5–50 μl), lithium dodecyl sulfate (LDS; 1: 4), and DTT (Dithiotreitol; 1:10) in ddH$_2$O. Samples were then heated at 95°C for 10 min. Next, samples were run on a polyacrylamide gel (bis-Tris 4–12%; Life Technologies), according to manufacturers' recommendations. Migration buffer was made freshly (20× MOPS buffer, 1:20; Life Technologies) in ddH$_2$O. Samples were run for 90 min at 100V and 40 mA. After migration, proteins were transferred to a polyvinylidene fluoride membrane (PVDF) activated with 100% ethanol. The transfer was carried out in a transfer tank filled with transfer buffer (ethanol 1:10 in 1× transfer buffer; Life Technologies) at 30V, 400 mA for 1 h. Then, membranes were rinsed in PBS-T (Tween 20 0.05% in PBS 1×) and incubated in blocking buffer for 1 h (Tween 20, 0.05% in 5% milk PBS). Membranes were then incubated at 4°C overnight with primary antibody solutions prepared in 5% milk in PBS-T, anti-lamin A/C (1:100, Sc6215; Santa Cruz), and anti-progerin (1:100, ab66587; Abcam). Membranes were washed in PBS-T (4 × 5 min) and then incubated 1 h at RT with secondary antibodies against appropriate species Alexa Fluor 488-555 (1:1,000; Abcam).

Last, membranes were washed in PBS-T (4 × 5 min), and pictures were taken using a BioRad Chemidoc Imaging System. Normalizations and quantifications were performed using the ImageJ software which calculates the ratio between the fluorescence of the loading control and the protein of interest, after background extraction.

### RNA extraction and RT-qPCR

RNA extraction was realized using the QIAGEN's RNeasy mini kit. Briefly, reverse transcription of 1 μg of total RNA was performed using the Superscript III kit and oligo dT following manufacturer's instructions at 42°C for 50 min followed by inactivation at 70°C for 15 min (Life Technologies). Primers were designed using Primer Blast. PCR amplification was performed on a LightCycler 480 (Roche) using the SYBR green master mix with the following program: preincubation at 95°C for 10 min then 40 cycles amplification each corresponding to 15 s at 95°C followed by 1 min at 60°C. The program ends with a merge step including a step of a second at 98°C, 30 s at 70°C and finally 10 s at 98°C.

Crossing-threshold (Ct) values were normalized by subtracting the geometric mean of three housekeeping genes (*GAPDH*, *PPIA*, and *HPRT1*). All Ct values were corrected by their PCR efficiency, determined by 1:2 or 1:4 cDNA dilution series. Results were treated with the GraphPad software (prism V8.0) for statistical tests (ANOVA, Kruskal–Wallis multiple comparison test; α set at 0.05). Only *P*-values less than 0.05 were considered as statistically significant. All analyses were carried out in biological duplicates and technical duplicates. Primer sequences are provided as supplemental information.

### RNA sequencing

After extraction, RNA quality was controlled by a bioanalyzer and quantified using QuBit. RNA sample with a RIN >8 was considered for sequencing with 1 μg used per condition. For sequencing the universal plus, mRNA-seq from NuGEN was used for preparation of poly-A RNAs libraries. The same bioinformatics analysis was applied for both datasets of RNA-Seq generated from two sequencing cores: Integragen using TruSeq Stranded Total RNA-Illumina/Illumina HiSeq 4000 (2 × 75 bp) (2017) and GBiM using mRNA universal Nugen-Tecansur kit/Illumina NextSeq 500 (2 × 75 bp) (2018). A total of 39 samples were analyzed. Sequencing reads were aligned to the Human genome reference GRCh38 using STAR v2.5.3a (69). Files indexes were generated with Sambamba (v0.6.6), and aligned reads on genes (GENCODE) were quantified using Stringtie (v1.3.1c). Finally, DEGs between conditions were identified with the R package DESeq2 (v1.18.1). Overrepresentation test analyses were performed using enrichGO from the R package clusterProfiler (v3.10.1) (70). DEG identified in RNA-seq with an FDR-adjusted *P*-value < 0.05 and abs($\log_2$FC) > 2 were used as input. Identified GO terms in biological process ontology were selected based on an FDR-adjusted *P*-value < 0.05. Heatmaps were obtained thanks to the pheatmap (v1.0.12) R package using TPM values.

RNA-seq data and raw count matrix were deposited at the NCBI Gene Expression Omnibus (https://www.ncbi.nlm.nih.gov/geo/) under the complete SuperSeries accession number GSE202369 (RNASes, GSE202364; EPIC Array 850K, GSE202368).

### Infinium MethylationEPIC array

Genome-wide DNA methylation analysis was performed by Infinium MethylationEPIC Array through Diagenode services. Genomic DNA was extracted using the NucleoSpin Tissue kit (Macherey-Nagel) from two different cell pellets for each sample. A minimum of 500 ng was sent to Diagenode services for DNA methylation analysis. The analysis was mainly carried out using the R package ChAMP (71). Probes with missing values are removed, samples with more than 10% of probes with a detection *P*-value greater than 0.01, probes with a detection *P*-value greater than 0.01 in at least one sample, and probes for which 5% of samples have a bead count less than three are filtered out. An annotation file is loaded that contains information about the location of probes—such as chromosome, position, and nearby genes (72). Probes targeting CpG sites that are near SNPs, belong to X/Y chromosomes, or align with multiple locations are filtered out. A matrix of methylation β values is returned. The β-value indicates the percentage of copies of a CpG site from a given sample that were methylated (73) determined for each CpG location as the relative intensity of methylated signal (M) and unmethylated signal (U) (35).

Samples from the different groups were compared with identify DMPs based on a significantly different average methylation level at CpG site. The ChAMP function for identifying DMPs uses the limma package (74). A 0.2 absolute difference in mean β values (Δβ) was used to identify DMPs (75). The ComBat method was used to correct for batch effects (74). Heatmaps were realized using the pheatmap (v1.0.12) R package using β value matrix for methylation. Violin plots were realized by converting β-value matrix to M-value (for better visualization) using the lumi (v2.42.0) R package then by plotting values using the ggplot2 (v3.3.3) R package. DMPs with an FDR-adjusted *P*-value < 0.05 and an abs(Δβ) > 0.2 were represented as bar plots using the ggplot2 (v3.3.3) R package for genomic features and CGI status according to Illumina EPIC array annotation file. Comparisons were made relative to the REMC (Roadmap Epigenomics Mapping Consortium) features for EO25 adipose-derived MSC cultured cells. Corrected betavalue matrix along with closest distance to lamin A LADs (GSM1313397) for annotated probes in EPIC array were used to create 2D density plot using ggplot2 (v3.3.3) R package.

## Availability of Data and Material

All data supporting the findings of this study are available; further information is available within the supplementary section of this manuscript. Materials are available upon request.

### Ethics approval and consent to participate

Parents have provided written informed consent for the use of biopsies for medical research in accordance with the Declaration of Helsinki. Samples were provided by the Center for Biological Resources (Department of Medical Genetics, La Timone Children's Hospital) with the AC 2011-1312 and No IE-2013-710 accreditation numbers.

## Supplementary Information

## Acknowledgements

We are indebted and thank all patients for participating in this study. Source of funding: This study was funded by "Association Française contre les Myopathies," AFM Telethon (AFM; TRIM-RD), Aix Marseille University Initiative of Excellence (A*Midex, VinTAge project), and the INSERM-Agemed cross-cutting program. K Annab was the recipient of a fellowship from the French Ministry of Education and Fondation pour la Recherche Médicale (FDT 20170437114). Study sponsors had no role in study design, collection analysis, data interpretation, writing of the report, or decision to submit the paper for publication.

### Author Contributions

JP Trani: data curation, formal analysis, validation, investigation, and writing—review and editing.
R Chevalier: data curation, software, formal analysis, validation, and writing—review and editing.
L Caron: data curation, formal analysis, validation, investigation, and writing—review and editing.
C El Yazidi: validation and methodology.
N Broucqsault: validation and methodology.
L Toury: formal analysis.
M Thomas: validation and methodology.
K Annab: validation, investigation, and methodology.
B Binetruy: conceptualization, funding acquisition, and writing—review and editing.
A De Sandre-Giovannoli: resources and writing—review and editing.
N Levy: resources and funding acquisition.
F Magdinier: conceptualization, data curation, formal analysis, supervision, funding acquisition, project administration, and writing—review and editing.
JD Robin: conceptualization, data curation, formal analysis, supervision, validation, investigation, methodology, project administration, and writing—original draft, review, and editing.

### Conflict of Interest Statement

The authors declare that they have no conflict of interest.

### Author Contribution

JP Trani performed most of the experiments presented in the report and analyzed the data.
RC performed the bioinformatics data analysis.
N Broucqsault, L Caron, LT, and K Annab set up the differentiation assay and performed some experiments.
C El Yazidi and M Thomas produced or provided pluripotent stem cells.
B Binetruy participated in the study design.
A De Sandre-Giovannoli provided patient's samples, designed the study, and contributed to the edition of the manuscript.
N Levy provided patient's samples, designed the study, and obtained funding.
FM supervised the study, analyzed the data, wrote the manuscript, and obtained funding.
JDR performed some of the experiments, analyzed the data, and wrote and edited the manuscript.

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
