## [Reviewer comments · Life Science Alliance]

Life Science Alliance

Mesenchymal stem cells derived from patients with aging syndromes display hallmarks of aging

Jean Philippe Trani, Raphaël Chevalier, Leslie Caron, Claire El Yazidi, Natacha Broucqsault, Léa Toury, Morgane Thomas, Karima Annab, Bernard Binetruy, Annachiara De Sandre-Giovannoli, Nicolas Levy, Frederique Magdinier, and Jerome Robin
DOI: <https://doi.org/10.26508/lsa.202201501>

Corresponding author(s): Jerome Robin, Aix Marseille Univ and Frederique Magdinier, Aix-Marseille University

Review Timeline:	Submission Date:	2022-04-25
	Editorial Decision:	2022-06-03
	Revision Received:	2022-08-05
	Editorial Decision:	2022-08-26
	Revision Received:	2022-08-31
	Accepted:	2022-08-31

Scientific Editor: Novella Guidi

Transaction Report:

June 3, 2022

Re: Life Science Alliance manuscript #LSA-2022-01501

Dr. Jerome Robin
Aix Marseille Univ
MMG, Marseille Medical Genetics U1251
27 Bd Jean Moulin
Faculté de Médecine Campus Timone
Marseille 13385
France

Dear Dr. Robin,

Thank you for submitting your manuscript entitled "Mesenchymal stem cells derived from patients with aging syndromes display hallmarks of aging" to Life Science Alliance. The manuscript was assessed by expert reviewers, whose comments are appended to this letter. We invite you to submit a revised manuscript addressing the Reviewer comments.

Thank you for this interesting contribution to Life Science Alliance. We are looking forward to receiving your revised manuscript.

Sincerely,

B. MANUSCRIPT ORGANIZATION AND FORMATTING:

Reviewer #1 (Comments to the Authors (Required)):

The manuscript provides new insights into the pathogenesis of LMNA-linked progeroid syndromes, severe diseases that represent a model of the ageing process. Thus, the study is relevant to a broad readership. Importantly, the study helps understanding of HGPS and APS pathogenesis.

By analysing the transcriptome and methylome of hiPSC-derived MSCs, the authors identified disease-specific DNA methylation patterns and transcriptional profiles at early stages of MSC differentiation.

I SUGGEST TO ADD SOME HYPOTHESIS ON THE RELEVANCE OF DIFFERENTIAL DNA METHYLATION IN APS VERSUS HGPS, AS WELL AS ON THE INTERESTING DIFFERENCES DETECTED WHEN USING YOUNG DONOR CELLS OR AGED DONOR CELLS AS CONTROLS.

Interestingly, the authors further show alterations of mitochondrial pathways and hyperactivated DNA damage and telomere response pathways in patient-derived hiPSC-differentiated MSCs.

A KEY POINT THAT MUST BE DISCUSSED IS THAT ALL DNA-DAMAGE RELATED MARKERS HAVE BEEN EXPLORED UNDER BASAL CONDITIONS. FURTHER STUDIES ARE NEEDED TO EXPLORE THE SIGNIFICANCE OF ALTERED PATHWAYS WHEN DNA DAMAGING CONDITIONS OR STRESSORS ARE APPLIED TO CELLS.

ALONG THIS LINE, DIFFERENCES BETWEEN APS AND HGPS FORMS MIGHT BE LESS EVIDENT (OR EVEN MORE EVIDENT) UNDER CERTAIN STRESS CONDITIONS. THIS HYPOTHESIS MUST BE MENTIONED IN THE MANUSCRIPT.

Reviewer #2 (Comments to the Authors (Required)):

In the present manuscript, Trani et al. investigate the use of MSCs differentiated from iPSCs of patients suffering from premature ageing phenotypes (e.g. HGPS, APS) as a model for ageing research. They focus on transcriptomics and DNA methylation analysis in the iPSCs and MSCs. In general, the data is of high quality and there some interesting aspects in the study. However, I do have several concerns: The functional consequences of lamin mutations in tissue culture have been excessively explored over the last years. What is the advantage of using MSCs and what is the new insight provided here - this is something the authors should focus on more - what can we learn in these MSCs? In other words, be much clearer with your hypothesis. The described phenotypes are very similar to the ones in fibroblasts (eg. mitochondrial dysfunction, PMID: 32046343 or DNA damage, PMID: 26079711). HGPS MSCs have also been characterised before with respect to DNA methylation (and eradication of DNA methylation signatures upon iPSCs) and differentiation potential. Given these limitations, the analysis and the resulting conclusions are still too preliminary and descriptive and I would encourage the authors to explore their dataset further in a revised version of the manuscript.

Specific points:

The HGPS iPSC line expresses pluripotent factors to a much lower extend. Could the authors comment on the behaviour of this line, e.g. differentiation potential. Is there an explanation for this difference compared to the other iPSCs? Given these differences, it is surprising that the transcriptome is remarkably similar. The authors should comment here.

One classical characteristic feature of MSCs that should always be checked is the differentiation potential into adipocytic, chondrogenic and osteogenic lineage. According to previous reports (e.g. PMID: 29476423) this should be possible and thus, must be included in the basic characterisation of the differentiated MSCs.

There is methylation data on aged MSCs (e.g. PMID: 28844656). How does the methylation profile measured here correspond to these data. The analysis currently presented in the manuscript is very descriptive. Where are differentially methylated sites, what genes are encoded at these loci - TSS or gene body methylation. As it stands currently, the analysis is not very helpful in understanding the impact of DNA methylation in the used MSCs.

Figs. 2E-G are not really ideally suited to understand the relationship between the different samples and how distinct they are. A PCA might be more appropriate and/or genes (in these categories) differentially regulated between the samples should be highlighted. In addition, the authors should comment on previous work (e.g. PMID: 25241740) that showed a clear rejuvenation event upon reprogramming - how come that here, y and o MSCs still carry such profound signatures of ageing?

Minor points:

RNA-seq: A PCA analysis or correlation analysis should be reported to be able to judge the reproducibility of the experiment

Reviewer #3 (Comments to the Authors (Required)):

In their manuscript "Mesenchymal stem cells derived from patients with premature aging syndromes display hallmarks of physiological aging" Trani et al showed alterations in the age related pathways during MSC differentiation in the human induced pluripotent stem cells derived from progeroid patient of 3 different subgroups of laminopathies from HGPS, HGPS-L and APS. Furthermore they also reported differences in the mitochondrial pattern, distinct methylation patterns and increased DNA damage response in these MSCs, irrespective of progerin accumulation leading to similar premature aging phenotype as that of HGPS hence indicating different converging pathways responsible for this.

Comments:

This manuscript describes potentially interesting results based on the molecular level of the classical HGPS disease, specially focusing on the dysregulation in the pathways observed at the early stages of MSC differentiation. However, the manuscript could gain from further sharpening of the interpretation of the results. The data is further solidified with the use of patient derived iPSC cells and transcriptomic analysis. Overall the work is well written and mostly the conclusions are supported with the data shown. However some concerns are mentioned below:

All experiments were performed using different types of patient cell lines HGPS (n=3), HGPS-L (n=1) and APS (n=3) and as controls young and healthy aged donors were used.

Supplementary Fig 1B expressing pluripotency and differentiation markers is not really clear. How is the data validated? Maybe figure should be better and clearly labelled.

Supplementary Fig 1C again is confusing. Maybe it is better that authors change the labelling of figure to IPS-HGPS, IPS-APS and IPS-HGPS-L instead of just writing HGPS, AP and HGPS-L as these are induced pluripotent cells and not just normal cells hence it leads to confusion.

What could be the reason for the low expression of the classical MSC markers CD73 and CD105 in HGPS-L cells as shown in Supp Fig 2A? Maybe this can be explained a bit in discussion.

In the upcoming figures for instance supplementary figure 2 and Figure 1C they now refer to hiPSC derived MSCs but the labelling remains the same like previous figures hence leads to confusion again as they are MSCs derived from iPSC and not just alone iPSCs. Would suggest to label the figure like done in Figure 2A (heatmap).

Supplementary Fig 3A and 3B Venn diagrams in the heading the labelling should be consistent like in the figure legends left is comparison to CT-Y and right is CT-A.

At P7 differentiation stage in MSCs previously the authors examined high proportion of Progerin expression in HGPS-L MSCs, but in the transcriptomic analysis they failed to identify and dysregulated pathways at this stage. Perhaps they can explain this point in relevance to quoting references by other authors.

Fig 1C the legend is misleading, the d35 isoform is labeled by a yellow arrow. Here the authors should comment why they don't find progerin in the cells of healthy old donor, as progerin has been detected in lysates from old donors in the past.

Fig 2E-G reveals the GO analysis of the differences in the pathways observed in the MSCs. Maybe it is better that instead of just showing the expression profiles of genes associated to aging-associated processes they can also confirm this by doing some wet lab techniques like western blotting to show it at protein level or also perform qPCR to show clearly at mRNA level these differences.

Fig 3A are these the methylation patterns of MSCs only or of hiPSC derived MSCs? If these are of MSCs alone then what do you think would be the changes in the hiPSCs methylation patterns of the subgroups of laminopathies? Would there be any differences when you compare them to each other?

Fig 4 and supplementary Fig 4 are highly confusing maybe they should be explained better in the text as well as in figure legends.

Fig 5 the mitochondrial analyses should be complemented with DHE/DCF staining and quantification.

Fig 5C, the yH2Ax foci staining in Ctrl's what cells are used the young or old ones? As there is only one panel for staining showed but in the quantification graphs there are two bars one for young and one for aged. Even if there are no differences maybe they should show staining in both the young and old cells as they have shown quantification for both. Also maybe as they are highlighting in title hallmarks of physiological aging perhaps it would make the data stronger if they can check in their hiPSC derived MSCs the expression of other stress markers like 53BP1.

MSC are acknowledged for their potential to regenerate damaged tissue due to their ability to terminally differentiate into a broad variety of cell types. Since this manuscript focuses on the hiPSC derived MSCs from premature aging syndromes and the authors have already shown that the growth is improved, have any other physiological improvement been seen in these cells which could perhaps also lead to a therapeutic strategy in future? Perhaps authors can discuss this. Investigation of the mechanisms of stem cell aging is essential for in vitro expansion of stem cells that can be employed in both basic and clinical research.

All the study has been solely performed in the in-vitro models of premature aging. It would be better if they can also link the findings to the in-vivo mouse models of HGPS and related laminopathies and mention it in their discussion. Maybe this can help in widening our understanding of organ aging.

As hallmarks of physiological ageing are being discussed in this publication and MSC from APS subgroup of laminopathy has shown similar phenotype to HGPS MSC irrespective of progerin accumulation. What do the authors think about checking cellular senescence in these cells? What about p53 marker?

The authors have focused on the DNA methylation in the manuscript and have shown expression profiles also but maybe they can also hint on other epigenetic alterations like histone modifications and chromatin remodeling and also non-coding RNAs. What do they think about transcriptional silencing in these cells?

The stem cell niche plays a critical role in maintaining the stemness properties and proper functioning of the stem cells. Was the expression of FGF (Fibroblast growth factor) checked throughout the experiments as it increases after aging and can also impair with the self-renewal of stem cells. Moreover inflammatory markers can also negatively affect stem cell function for this inflammation is considered as markers of aging. What are authors remark to this?

The discussion should focus more on carving out the common aberrant features of the analyzed patient/ old cells as this could help in guiding future studies on the pathophysiology of premature aging, for example discuss that overall methylation might not be relevant as it is not deviant in APS, whereas nuclear shape is a common feature of laminopathy cells as also mitochondrial aging.

Using genome-wide approaches on models from Laminopathies with or without progerin accumulation (HGPS, HGPS-L, APS), we provide new insights on pathways altered during early stages of mesenchymal stem cells differentiation.

We thank the reviewers for their in-depth constructive criticisms of our work. We provide below a complete response to all points raised and suggestions made. Modification to the manuscript are found below and have been added to the updated version of our manuscript. We believe the changes made are improving the clarity and quality of our study.

Reviewer #1

The manuscript provides new insights into the pathogenesis of LMNA-linked progeroid syndromes, severe diseases that represent a model of the ageing process. Thus, the study is relevant to a broad readership. Importantly, the study helps understanding of HGPS and APS pathogenesis. By analysing the transcriptome and methylome of hiPSC-derived MSCs, the authors identified disease-specific DNA methylation patterns and transcriptional profiles at early stages of MSC differentiation.

I suggest to add some hypothesis on the relevance of differential DNA methylation in APS vs. HGPS, as well as on the interesting differences detected when using young donor or aged donor cells as controls.

We thank the reviewer for this justified remark.

While reprogramming can reset the epigenetic aging clock, it has been demonstrated that a significant residual epigenetic signature of the donor age remains in hiPSC (PMID: 27941802). HiPSC derived from older donors exhibit different DNA methylation pattern compared to hiPSC derived from younger donor. This residual age-associated signature is most likely carried on upon differentiation (in our case into MSCs) and therefore conserved in hiPSC-derived MSCs, explaining the differences in methylation profiles observed between our young and old donors control cell lines. This is now being discussed in our revised manuscript.

“While reprogramming can reset the epigenetic aging clock, it has been demonstrated that a significant residual signature of the donor age remains in hiPSCs (PMID: 27941802), while most can be erased by extensive hiPSCs expansion along with an increased risk of genetic alteration and epimutations (PMID : 31143763). An overall resistance to demethylation has been observed in hiPSCs from elderly donors; in accordance with this, age-related CpGs in hiPSCs derived from older donors display increased methylation levels compared to hiPSC

derived from young individuals. Because we kept our hiPSCs at low passages, this residual age-associated signature is most likely carried on in our hiPSCs, conserved upon differentiation into hiPSC-derived MSCs and explains the differences in methylation profiles observed between our young and old donors control cell lines.”

In agreement, this could allow one to distinguish between healthy aging situations and specificity of pathological accelerated aging as markedly observed in HGPS and HGPS-L. We have updated our discussion accordingly.

“Further, our work report common aberrant features between pathologies and hiPSC-derived MSCs from an old donor with the exclusion of DNA methylation. Indeed, hiPSC-derived MSCs from APS patients and the young donor share similar DNA methylation pattern suggesting that DNA methylation might not be a key factor in premature ageing. This observation is complementary to previous work (*PMID: 25241740*) but requires a higher number of samples (we report 7 *LMNA*-associated pathologies and 2 controls) before additional speculations, due to potential compensations in APS cells for example.”

Interestingly, the authors further show alterations of mitochondrial pathways and hyperactivated DNA damage and telomere response pathways in patient-derived hiPSC-differentiated MSCs. A key point that must be discussed is that all DNA-damage related markers have been explored under basal conditions.

As suggested, we have updated the manuscript to include this pertinent observation in our discussion.

“Importantly, all our observations (nuclear shapes, DNA damage, mitochondrial network) are made under basal conditions. Further studies are needed to fully grasp changes detected across pathologies. This include the use of stressor and protector (DNA damage) or ROS and antioxidant (mitochondrial network). Depending on their associate *LMNA* mutations, differences observed could be erased (due to resilience) or exacerbated.”

Our work aimed to identify the first signs appearing upon differentiation, in an unchallenged protocol. As the reviewer point out, challenged conditions (UV, oxidative stress, serum deprivation) could reveal a different outcome depending on the resilience of our models. We believed however that this is beyond the scope of the present study

Reviewer #2

In the present manuscript, Trani et al. investigate the use of MSCs differentiated from iPSCs of patients suffering from premature ageing phenotypes (e.g. HGPS, APS) as a model for ageing research. They focus on transcriptomics and DNA methylation analysis in the iPSCs and MSCs. In general, the data is of high quality and there some interesting aspects in the study.

However, I do have several concerns: The functional consequences of lamin mutations in tissue culture have been excessively explored over the last years. What is the advantage of using MSCs and what is the new insight provided here - this is something the authors should focus on more - what can we learn in these MSCs? In other words, be much clearer with your hypothesis.

We thank the reviewer for these comments. Our study, to our knowledge, is the first to report the transcriptome and methylome analyses of multiple rare Laminopathies,

including APS samples. Lamin mutations have been investigated in a number of models but never in the perspective of differentiation and establishment of anomalies associated with the differentiation process and re-expression of A-type Lamins. Our work aimed to fill that gap by also including samples of patients sharing the clinical phenotype but without progerin accumulation (APS). Previous works have mostly focused on fibroblasts or a stand point of hiPSCs, all of which never included APS individuals. Regarding the clinical phenotype and tissues impacted in progeroid patients, MSCs represent the first cells in development that produce LMNA and hold the potential to differentiate in cells from all tissues affected.

We have updated the introduction of our revised manuscript accordingly to better state our approach.

“Importantly, hiPSCs hold the potential to produce standardized cell preparation (Froebel et al., 2014) with large capacity of culture expansion. Thus, we produced hiPSC-derived MSCs and focused on events associated with early LMNA expression (e.g., Lamins and Progerin), including transcription, DNA methylation and major cellular hallmark of aging (DNA damage, mitochondrial network).”

The described phenotypes are very similar to the ones in fibroblasts (eg. mitochondrial dysfunction, PMID: 32046343 or DNA damage, PMID: 26079711). HGPS MSCs have also been characterised before with respect to DNA methylation (and eradication of DNA methylation signatures upon iPSCs) and differentiation potential. Given these limitations, the analysis and the resulting conclusions are still too preliminary and descriptive and I would encourage the authors to explore their dataset further in a revised version of the manuscript.

We understand the point raised by this reviewer. However, our strategy was to investigate the early events leading to potential defects in cells isolated from patients carrying LMNA mutations. We were surprised to observe the fast establishment of the Lamin-associated phenotypes that mimic earlier observations made on fibroblasts. Aside from cell type differences (we used hiPSCs and differentiated MSCs), our work gather samples carrying different rare LMNA mutations, for a total of 7 individuals with LMNA mutations, a panel or in depth analysis not reached previously.

Our results confirm the relevance of previous observations made in primary fibroblasts. Besides, we showed that if hiPSCs do not exhibit signs of laminopathies, induction of LMNA expression as the cells differentiate re-established classical cellular signs. A crucial point, showing that defects are installed at early stages, upon progerin detection by WB.

Specific points:

The HGPS iPSC line expresses pluripotent factors to a much lower extend. Could the authors comment on the behaviour of this line, e.g. differentiation potential. Is there an explanation for this difference compared to the other iPSCs? Given these differences, it is surprising that the transcriptome is remarkably similar. The authors should comment here.

As shown in the updated Sup Figure 1A, the three HGPS-hiPSC lines express lower levels of pluripotent factors than hESC. However, we did not observe differentiation defects in any of these hiPSC lines, which retain their capacities to differentiate into all three germ layers as demonstrated by the differentiation markers expression upon Embryoid Bodies formation (Updated Sup Figure 1B-C).

As suggested, we have updated our manuscript by including a remark in our results section, commenting this specific point.

“To note, while HGPS-hiPSC lines expressed lower levels of pluripotency markers, we did not observe any differentiation defects in these lines.”

Further, we have modified the figures associated to Supplement 1 with a more detailed analysis of hiPSCs and their differentiation potential.

One classical characteristic feature of MSCs that should always be checked is the differentiation potential into adipocytic, chondrogenic and osteogenic lineage. According to previous reports (e.g. PMID: 29476423) this should be possible and thus, must be included in the basic characterisation of the differentiated MSCs.

Our study focuses on the early stages of MSCs after hiPSCs reprogramming (9 cell line in total). As such we have included the characterization of these new hiPSCs lines (pluripotency markers) as well as the characterization of the MSCs derived (MSCs markers and absence of markers from the hematopoietic lineage). We believe looking into the final differentiation stage of MSCs in the three lineages to be out of the scope of this work.

However, we have checked their capacities to reach these final differentiations stages in each cell lines. All hiPSC-MSC from all individuals were differentiated successfully, as illustrated by the representative pictures included in this response. We are currently investigating these cell lines in a separate study focusing on the osteogenic lineage.

Ability of MSCs to differentiate into Osteoblasts (Alizarine red coloration), Adipocytes (oil red O coloration) and Chondrocytes was tested for all MSCs produced as a proof of concept. We report a representative picture of each differentiation assessed in one HGPS clone.

There is methylation data on aged MSCs (e.g. PMID: 28844656). How does the methylation profile measured here correspond to these data. The analysis currently

presented in the manuscript is very descriptive. Where are differentially methylated sites, what genes are encoded at these loci - TSS or gene body methylation. As it stands currently, the analysis is not very helpful in understanding the impact of DNA methylation in the used MSCs.

Taking these remarks into accounts, we retrieved the methylome data from this study (PMID 28844656) and compared it to our complete dataset. We first like to stressed out, that the data provided are from MSCs derived from bone marrow, our dataset is produced from MSCs derived from hiPSCs. Notable deviations are hence expected. Using 10 samples, this report elegantly described between changes in young vs. old MSCs but also from the changes induced by cell culture passages.

Using an Upset plot, we did not see a clear overlap with our data. One could expect the most overlap would be between our comparison Control Young vs. Control Aged (to replicate data provided by the study). Nevertheless, no overlap can be detected throughout the different dataset, due to a very low number of DMRs.

One hypothesis can be due to the nature of the cells (Bone marrow derived vs. hiPSCs derived) and methods used (ERRBS followed by WGBS vs. EpicArray of 850K CpGs) to define DMRs positions.

To note, from this work (Pasumarthy et al.), only DMRs identified between old vs young that overlap with TSS in a subset of genes could be easily retrieved. Indeed, the remaining data accessible is the complete untreated, unmapped set of NGS under SRA access (not treatable on our side).

Upset plot of DMRs found by Epic array compared to DMRs retrieved from a Whole genome study in old vs. young MSCs (PMID: 28844656).

In our updated manuscript, we provide the full analysis of methylation using ChroHMM and other approaches to get insight into functional elements. All complete and comprehensive results are given in Tables, with detailed legends and highlighted key numbers. Similar to the observation made by Pasumarthy and colleagues, we found that differential methylation is enriched at distal transcription factor binding sites (TSS). In agreement, we have modified our manuscript and include this pertinent reference.

Figs. 2E-G are not really ideally suited to understand the relationship between the different samples and how distinct they are. A PCA might be more appropriate and/or genes (in these categories) differentially regulated between the samples should be highlighted.

PCA results are provided in this response below using similar gene selection. Unlike HeatMaps, PCA will give an overall relationship but not a notion on the number of genes involved nor their orientation (up- or down- regulation). Thus, to keep our manuscript fluid and comprehensive to the largest audience, PCAs have not been included but provided for our rebuttal.

PCA analysis of DEGs in MSCs derived from hIPSCs associated with Telomere maintenance, DNA damage or Mitochondria, respectively.

In addition, the authors should comment on previous work (e.g. PMID: 25241740) that showed a clear rejuvenation event upon reprogramming - how come that here, y and o MSCs still carry such profound signatures of ageing?

While reprogramming can reset the epigenetic aging clock, it has been demonstrated that a significant residual signature of the donor age remains in hiPSC (PMID: 27941802). An overall resistance to demethylation has been observed in hiPSC from elderly donors and in accordance with this, age related CpG sites in hiPSC derived from older donors display increased methylation levels compared to hiPSC derived from young individuals. This residual age-associated signature is most likely carried on upon differentiation and therefore conserved in our hiPSC-derived MSCs. This is discussed in the revised version of the manuscript.

“While reprogramming can reset the epigenetic aging clock, it has been demonstrated that a significant residual signature of the donor age remains in hiPSCs (PMID: 27941802), while most can be erased by extensive hiPSCs expansion along with an increased risk of genetic alteration and epimutations (PMID : 31143763). An overall resistance to demethylation has been observed in hiPSCs from elderly donors; in accordance with this, age-related CpGs in hiPSCs derived from older donors display increased methylation levels compared to hiPSC derived from young individuals. Because we kept our hiPSCs at low passages, this residual age-associated signature is most likely carried on in our hiPSCs, conserved upon differentiation into hiPSC-derived MSCs and explains the differences in methylation profiles observed between our young and old donors control cell lines.”

It is to be noted that most of this residual age-related aberrant methylation signature can be erased by extended passaging. However, as explained in the discussion of our manuscript, for our experiments we used hiPSC at low passages to avoid genomic aberrations acquired during long expansion and that could potentially induce a drift unrelated to their disease-associated mutations.

“Importantly, we set our exploration in hiPSCs with low passages, as the persistent maintenance of hiPSCs might induce a drift unrelated to their associated mutations [46,47]. Likewise, all MSCs were analyzed at P7, the earliest passage where Progerin and isoform (Lamin Δ 35) could robustly be detected”.

Reviewer #3

In their manuscript "Mesenchymal stem cells derived from patients with premature aging syndromes display hallmarks of physiological aging" Trani et al showed alterations in the age related pathways during MSC differentiation in the human induced pluripotent stem cells derived from progeroid patient of 3 different subgroups of laminopathies from HGPS, HGPS-L and APS. Furthermore they also reported differences in the mitochondrial pattern, distinct methylation patterns and increased DNA damage response in these MSCs, irrespective of progerin accumulation leading to similar premature aging phenotype as that of HGPS hence indicating different converging pathways responsible for this.

Comments:

This manuscript describes potentially interesting results based on the molecular level of the classical HGPS disease, specially focusing on the dysregulation in the pathways observed at the early stages of MSC differentiation. However, the manuscript could gain from further sharpening of the interpretation of the results. The data is further solidified with the use of patient derived iPSC cells and transcriptomic analysis. Overall the work is well written and mostly the conclusions are supported with the data shown.

We thank this reviewer for the extensive review and constructive remarks. After reading all comments, we realized that some confusion could have been drawn by the way of our writing or how or panels were labeled. We have modified all figures in the updated version of the manuscript to carefully avoid any misinterpretation. Importantly, we had the impression that reviewer 3 understood that a mix of cells and cell types were used in our work. However, we present here results from newly reprogrammed hiPSCs and their differentiation into MSCs. Primary cells available for progeroid patients are usually fibroblasts. Thus, we first derived hiPSCs and further differentiated them into MSCs. We have updated our text to make this point clear throughout this updated version.

“After optimization of the mesenchymal cell lineage differentiation from hiPSCs, hiPSC-derived MSCs, dubbed thereafter MSCs [25] (Figure 1B)”

However some concerns are mentioned below:

All experiments were performed using different types of patient cell lines HGPS (n=3), HGPS-L (n=1) and APS (n=3) and as controls young and healthy aged donors were used. Supplementary Fig 1B expressing pluripotency and differentiation markers is not really clear. How is the data validated? Maybe figure should be better and clearly labelled.

Following advice from two of the reviewers, we have updated this figure and present now the full report validating the pluripotency of cells in the different cell lines and patients.

Supplementary Fig 1C again is confusing. Maybe it is better that authors change the labelling of figure to IPS-HGPS, IPS-APS and IPS-HGPS-L instead of just writing HGPS, AP and HGPS-L as these are induced pluripotent cells and not just normal cells hence it leads to confusion.

Figure has been updated to reflect the information also provided in the legends.

What could be the reason for the low expression of the classical MSC markers CD73 and CD105 in HGPS-L cells as shown in Supp Fig 2A? Maybe this can be explained a bit in discussion.

We have updated the results to include a comment towards this lower expression. We do not have a clear hypothesis, as we only observed this in this particular cell-line. One could imagine that this is restricted to this clone, or that a link to the HGPS-L pathology can be made. With only one individual to explore, we feel this is a premature statement to make.

“HGPS-L clones displayed a steady medium percentage of positive cells (50%) for the CD73 and CD105 markers whereas no differences were observed for CD90 (>95% of positive cells), without inducing detectable defects in MSCs properties. This intriguing result should be explored in additional hiPSCs from HGPS-L individuals.”

In the upcoming figures for instance supplementary figure 2 and Figure 1C they now refer to hiPSC derived MSCs but the labelling remains the same like previous figures hence leads to confusion again as they are MSCs derived from iPSC and not just alone iPSCs. Would suggest to label the figure like done in Figure 2A (heatmap). Supplementary Fig 3A and 3B Venn diagrams in the heading the labelling should be consistent like in the figure legends left is comparison to CT-Y and right is CT-A. At P7 differentiation stage in MSCs previously the authors examined high proportion of Progerin expression in HGPS-L MSCs, but in the transcriptomic analysis they failed to identify and dysregulated pathways at this stage. Perhaps they can explain this point in relevance to quoting references by other authors.

Figure labels have been updated. RNASeq does not allow to detect progerin (produced from a post-translational defect, hence assessed rigorously only by WB). To speculate on RNASeq data, one could assume that transcripts would vary in cells with and without LMNA mutations (or producing Progerin and associated mutation). Hence, if one look at TPM (Transcripts Per Kilobase Million) of LMNA. We provide the table below as an indication:

	Mean TPM LMNA
MSC HGPS	38.4
MSC HGPSI	27.3
MSC APS	72
MSC CT-Y	92.5
MSC CT-A	68.3
hiPSC HGPS	10.35
hiPSC HGPSI	12.2
hiPSC APS	12.3
hiPSC CT-Y	7.7
hiPSC CT-A	10.9

By deduction, one can observed that 1/ LMNA transcripts are rare in all our hiPSCs (as reported in our manuscript); 2/ The CT-Y is the samples that displays the highest TPM along with APS, as expected; 3/ HGPS and HGPS-L present the lowest TPM, suggesting alternative transcripts; 4/ in CT-A we report a medium TPM, that probably hint to the presence of the production of Progerin, as suggested by previous studies in old donors. However, if one wants to detect Progerin from transcriptomic data, it needs to be on a long-read format to avoid any conjectural hypothesis and observations.

Fig 1C the legend is misleading, the d35 isoform is labeled by a yellow arrow. Here the authors should comment why they don't find progerin in the cells of healthy old donor, as progerin has been detected in lysates from old donors in the past.

Fig 1C report the WB performed in MSCs derived from hiPSCs. Thus, healthy old controls have been reprogrammed and we did not detect progerin upon MSCs differentiation. Even if the reprogramming in hiPSCs do not erased all hallmarks of aging (as observed in our report upon differentiation), progerin does not seem to be one of those. Differences observed in healthy old donor might be due to a potential low expression of progerin in few nuclei; an incomplete methylation pattern erasure, or both. A low progerin expression could elegantly explain the similitudes observed between Healthy old and HGPS; HGPS-L MSCs hallmarks.

Fig 2E-G reveals the GO analysis of the differences in the pathways observed in the MSCs. Maybe it is better that instead of just showing the expression profiles of genes associated to aging-associated processes they can also confirm this by doing some wet lab techniques like western blotting to show it at protein level or also perform qPCR to show clearly at mRNA level these differences.

In agreement with this comment we have performed complementary RT-qPCR experiments. They have been added to the supplemental figures of the updated manuscript (Supplemental Figure 5). In addition of representative genes of pathways including; DNA repair, Mesenchyme development, cartilage development, heart development and the growth signaling cascade, we report analyses of genes involved globally in Aging (e.g., *FOXO3A*, *APOE*, *MMP2*, *MMP9*, *MT-ND1*); provided below.

Relative gene expression in MSCs derived from hiPSCs of patients with LMNA mutations at P7 of genes involved in Ageing. Expression of selected genes is normalized to housekeeping genes (*HKG*; *PPIA*, *HPRT*, *GAPDH*) and control (young). For each condition, we report the average of biological and technical duplicates. Kruskal-Wallis test p-value * <0.05; ** <0.005; *** <0.0005; **** <0.0001.

We did not include these in our manuscript as we feel it cuts from the fluidity and clarity of the text. Additional qPCRs of genes involved in different pathways might not be more informative regarding the scope of our study focusing on pathologies associated to different *LMNA* variants and not ageing, *per se*.

Fig 3A are these the methylation patterns of MSCs only or of hiPSC derived MSCs? If these are of MSCs alone then what do you think would be the changes in the hiPSCs methylation patterns of the subgroups of laminopathies? Would there be any differences when you compare them to each other?

We thank the reviewer for making this remark as we realized that our explanation on the cellular model used was not sufficiently detailed. In this work, we only use cells derived from hiPSCs. We have updated the manuscript to avoid any confusion.

Fig 4 and supplementary Fig 4 are highly confusing maybe they should be explained better in the text as well as in figure legends.

Figures 4 (with supplemental) visually represents the methylation pattern at LADs (fig 4) and their borders (sup 4). We have updated the legends to ensure that readers can grasp the information provided. A simple table, or a curve (represented in the right panel) is not as informative as the heatmap given. We feel this representation is the best suited for the point raised in our text.

Fig 5 the mitochondrial analyses should be complemented with DHE/DCF staining and quantification.

Our work focus on early event observed concomitant to expression of the *LMNA* gene. Thus, we analyzed DNA damage, nuclear shapes, telomere and mitochondrial network. All in unchallenged conditions. We understand the point raised in this comment. However, DHE/DCF staining is beyond the scope of this study. DHE/DCF is an indicator of ROS formation, which could reflect mitochondrial function defects. As widely accepted, if ROS and oxidation are evaluated, a numerous number of assays needs to be performed to be informative. Further experiments are required to assess properly the ROS and oxidative situation of these cells with and without challenged conditions (as for the DNA damage). We have updated the manuscript to include this comment in the discussion.

“Importantly, all our observations (nuclear shapes, DNA damage, mitochondrial network) are made under basal conditions. Further studies are needed to fully grasp changes detected across pathologies. This include the use of stressor and protector (DNA damage) or ROS and antioxidant (mitochondrial network). Depending on their associate *LMNA* mutations, differences observed could be erased (due to resilience) or exacerbated.”

Fig 5C, the γ H2Ax foci staining in Ctrl what cells are used the young or old ones? As there is only one panel for staining showed but in the quantification graphs there are two bars one for young and one for aged. Even if there are no differences maybe they should show staining in both the young and old cells as they have shown quantification for both. Also maybe as they are highlighting in title hallmarks of

physiological aging perhaps it would make the data stronger if they can check in their hiPSC derived MSCs the expression of other stress markers like 53BP1.

Figure has been updated to include both Controls.

We did perform 53BP1 staining. However, we did not include the results in our manuscript for the sake of clarity and not over-presenting data that can be seen as redundant (DNA repair for H2AX; checkpoint / sensor of dsDNA break for 53BP1). Staining and quantification are presented below (can be added to the updated manuscript upon request from this reviewer).

MSCs from patients with premature ageing display an increase double strand break DNA damage response. A. 53BP1 immunostaining in hiPSC-derived MSC at passage 7 (green dots) for the different types of samples (Young healthy donor (CT-Y), aged healthy donor (CT-A), HGPS, Progeria like (HGPS-L) and atypical progeroid syndrome (APS). **B.** Quantification of the number of 53BP1 foci per nuclei. Experiments were carried out, in duplicate with counting of 200 nuclei. Holm-Sidak's multiple comparisons test, p-value * <0.0332 ; ** <0.0021 ; *** <0.0002 ; **** <0.0001 . Ladder bar respectively 30 μ m - 10 μ m.

MSC are acknowledged for their potential to regenerate damaged tissue due to their ability to terminally differentiate into a broad variety of cell types. Since this manuscript focuses on the hiPSC derived MSCs from premature aging syndromes and the authors have already shown that the growth is improved, have any other physiological improvement been seen in these cells which could perhaps also lead to a therapeutic strategy in future?

As reported above (see responses to reviewer#2), capacity of MSCs to reach final differentiation (osteogenic, adipogenic and chondrogenic differentiation) was assessed. We did not detect yet aberrant differences between conditions. We are currently exploiting these models (Osteoblasts/ Chondrocyte) in an ongoing study.

Perhaps authors can discuss this. Investigation of the mechanisms of stem cell aging is essential for in vitro expansion of stem cells that can be employed in both basic and clinical research. All the study has been solely performed in the in-vitro models of premature aging. It would be better if they can also link the findings to the in-vivo mouse models of HGPS and related laminopathies and mention it in their discussion. Maybe this can help in widening our understanding of organ aging.

We have updated the discussion of our manuscript to include a comment towards the use of our finding in research and link to studies for *in vivo* models.

"Further, our work report common aberrant features between pathologies and hiPSC-derived MSCs from an old donor with the exclusion of DNA methylation. Indeed, hiPSC-derived MSCs from APS patients and the young donor share similar DNA methylation pattern suggesting that DNA methylation might not be a key factor in premature ageing. This observation is complementary to previous work (PMID: 25241740) but requires a higher number of samples (we report 7 LMNA-associated pathologies and 2 controls) before additional speculations, due to potential compensations in APS cells for example. As recently performed in mice (cabral et al., 2021), future studies using our cellular model could assess current leading treatments for HGPS (e.g., farnesyl transferase inhibitor, rapamycin analog to promote clearance) and their potential effects in associated LMNA diseases (HGPS-L), including without Progerin accumulation (APS). Taking advantage of hiPSCs and their derived MSCs, treatment could be performed as early as possible in the differentiation process to appreciate if one can reverse modestly (or completely) hallmarks (nuclear abnormalities, DNA damage) as recently reported in a in vivo model."

As hallmarks of physiological ageing are being discussed in this publication and MSC from APS subgroup of laminopathy has shown similar phenotype to HGPS MSC irrespective of progerin accumulation. What do the authors think about checking cellular senescence in these cells? What about p53 marker?

We thank the reviewer for this judicious comment. Stemming from the data gathered by immunostaining (53BP1, H2AX) and deformation per nuclei, we observe that MSCs (derived from hiPSCs) from APS exhibit a striking phenotype similar to HGPS. As we did not detect significant differences in growth between these cells (trend to a faster growth in APS), we do not believe senescence is at play during the time of experiments. Additionally, as reported in our manuscript, no changes in replication rates could be detected across samples up to P12. This strongly advocates against a senescence effect in these cells.

The authors have focused on the DNA methylation in the manuscript and have shown expression profiles also but maybe they can also hint on other epigenetic alterations like histone modifications and chromatin remodeling and also non-coding RNAs. What do they think about transcriptional silencing in these cells?

The stem cell niche plays a critical role in maintaining the stemness properties and proper functioning of the stem cells.

Was the expression of FGF (Fibroblast growth factor) checked throughout the experiments as it increases after aging and can also impair with the self-renewal of stem cells. Moreover inflammatory markers can also negatively affect stem cell function for this inflammation is considered as markers of aging. What are authors remark to this?

Regarding FGF, it is important to note that it is the serum levels of FGF that is found increased in aging. Thus, cell type specificity is expected. In our study, we focused on MSCs derived from hiPSCs, FGF belongs to a very large family of growth factors (around 23 proteins from 23 genes – FGF1 to FGF23 identified so far). Nevertheless, we analyzed FGF21 expression by RT-qPCR, as the FGF member the most associated with aging. We observed a trend toward overexpression in MSCs derived from hiPSCs of the healthy aged donor. Regarding patients, this trend is only observed in APS individuals. Further experiments focusing on stemness and inflammation should be carried before making any conclusions.

FGF21

*Relative gene expression in MSCs derived from hiPSCs of patients with LMNA mutations at P7 of FGF21. Expression is normalized to housekeeping genes (HKG; PPIA, HPRT, GAPDH) and control (young). We report the average of biological and technical duplicates. Kruskal-Wallis test p-value * <0.05.*

To our knowledge, hiPSCs derived from all individuals were capable to differentiate into the three lineages without aberrant visual phenotypes. Results from RT-qPCR are provided in the updated supplemental (Supplemental Figure 5).

The discussion should focus more on carving out the common aberrant features of the analyzed patient/ old cells as this could help in guiding future studies on the pathophysiology of premature aging, for example discuss that overall methylation might not be relevant as it is not deviant in APS, whereas nuclear shape is a common feature of laminopathy cells as also mitochondrial aging.

We have updated the discussion of the manuscript. We have included specific points towards, hypothesis, future direction into clinics and the specific point raised by this reviewer on DNA methylation.

August 26, 2022

RE: Life Science Alliance Manuscript #LSA-2022-01501R

Dr. Jerome Robin
Aix Marseille Univ
MMG, Marseille Medical Genetics U1251
27 Bd Jean Moulin
Faculté de Médecine Campus Timone
Marseille 13385
France

Dear Dr. Robin,

Thank you for submitting your revised manuscript entitled "Mesenchymal stem cells derived from patients with aging syndromes display hallmarks of aging". We would be happy to publish your paper in Life Science Alliance pending final revisions necessary to meet our formatting guidelines.

- please add ORCID ID for secondary corresponding author-they should have received instructions on how to do so
- please add the Twitter handle of your host institute/organization as well as your own or/and one of the authors in our system
- please make sure the author order in your manuscript and our system match and that all authors listed in the manuscript file are also added to our system
- please add the supplementary figure legends at the end of the main manuscript text
- please add a complete Materials and Methods section to the manuscript

Figure Check:

- for Figure 1C, please indicate in legend if blots on the right are zoomed in versions of what is on the left
- please include a scale bar for the 3D reconstruction section of Figure 5E
- please include sizes next to all blots

A. FINAL FILES:

B. MANUSCRIPT ORGANIZATION AND FORMATTING:

Sincerely,

Reviewer #1 (Comments to the Authors (Required)):

The manuscript addresses interesting aspects of premature ageing, which help understanding pathomechanisms of progeroid laminopathies as well as common ageing processes. As requested, the authors have highlighted the limitations of the study, including differentiation towards MSC only and analysis performed in the absence of any stress stimulus. Many other aspects raised by the other referees have been discussed properly and a few changes have been added based on new experimental results.

Reviewer #2 (Comments to the Authors (Required)):

The authors have addressed all my main concerns and the hypothesis has been worked out much more clearly. I still believe that the addition of differentiation potential (as shown in the rebuttal) would present a useful validation of their experimental system and I encourage the authors to include those. Overall, I do support publication.

Reviewer #3 (Comments to the Authors (Required)):

the authors have revised the manuscript according to our and the other referee's comments satisfactorily. They addressed a lot of points experimentally and thereby really improved the article. we recommend the acceptance of the revised manuscript.

August 31, 2022

RE: Life Science Alliance Manuscript #LSA-2022-01501RR

Dr. Jerome Robin
Aix Marseille Univ
MMG, Marseille Medical Genetics U1251
27 Bd Jean Moulin
Faculté de Médecine Campus Timone
Marseille 13385
France

Dear Dr. Robin,

Thank you for submitting your Research Article entitled "Mesenchymal stem cells derived from patients with aging syndromes display hallmarks of aging". It is a pleasure to let you know that your manuscript is now accepted for publication in Life Science Alliance. Congratulations on this interesting work.

DISTRIBUTION OF MATERIALS:

Again, congratulations on a very nice paper. I hope you found the review process to be constructive and are pleased with how the manuscript was handled editorially. We look forward to future exciting submissions from your lab.

Sincerely,
